

# Hilbert space fragmentation and slow dynamics in particle-conserving quantum East models

**Pietro Brighi⋆, Marko Ljubotina and Maksym Serbyn**

IST Austria, Am Campus 1, 3400 Klosterneuburg, Austria

⋆ pbrighi@ist.ac.at

## Abstract

Quantum kinetically constrained models have recently attracted significant attention due to their anomalous dynamics and thermalization. In this work, we introduce a hitherto unexplored family of kinetically constrained models featuring conserved particle number and strong inversion-symmetry breaking due to facilitated hopping. We demonstrate that these models provide a generic example of so-called quantum Hilbert space fragmentation, that is manifested in disconnected sectors in the Hilbert space that are not apparent in the computational basis. Quantum Hilbert space fragmentation leads to an exponential in system size number of eigenstates with exactly zero entanglement entropy across several bipartite cuts. These eigenstates can be probed dynamically using quenches from simple initial product states. In addition, we study the particle spreading under unitary dynamics launched from the domain wall state, and find faster than diffusive dynamics at high particle densities, that crosses over into logarithmically slow relaxation at smaller densities. Using a classically simulable cellular automaton, we reproduce the logarithmic dynamics observed in the quantum case. Our work suggests that particle conserving constrained models with inversion symmetry breaking realize so far unexplored dynamical behavior and invite their further theoretical and experimental studies.

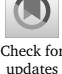

# 1 Introduction

In recent years, kinetically constrained models, originally introduced to describe classical glasses [1–3], have received considerable attention in the context of non-equilibrium quantum dynamics [4–8]. In analogy with their classical counterparts, they are characterized by unusual dynamical properties, including slow transport [7, 9–13], localization [14–16] and fractonic excitations [17, 18]. Additionally, in the quantum realm, other interesting phenomena have been observed, such as Hilbert space fragmentation [15, 19–25] and quantum many-body scars [26–30].

Among the many possible types of constraints, one can distinguish models that are inversion symmetric from those that break inversion symmetry. Among the latter models, the so-called quantum East model [8, 14, 31–34] where spin dynamics of a given site is facilitated by the presence of a particular spin configuration *on the left* represents one of the most studied examples. The quantum East model has been shown to host a localization-delocalization transition in its ground state [14], which allows the approximate construction of excited eigenstates in matrix product state form. Transport in particle-conserving analogues of the East model was recently investigated through the analysis of the dynamics of infinite-temperature correlations, revealing subdiffusive behavior. A similar result has also been observed in spin-1 projector Hamiltonians [35].

The interplay of particle conservation and kinetic constraints that break inversion symmetry opens several interesting avenues for further research. First, the phenomenon of so-called Hilbert space fragmentation that is known to occur in constrained models and is characterized by the emergence of exponentially many disconnected subsectors of the Hilbert space is expected to be modified. The additional $U(1)$ symmetry is expected to influence Hilbert space fragmentation beyond the picture presented in previously studied models [8, 14, 31]. Second, the presence of a conserved charge allows the study of transport [7, 12, 13]. While transport without restriction to a particular sector of fragmented Hilbert space results in slow subdiffusive dynamics [7, 12], a recent work [36] demonstrated that a restriction to a particular sector of fragmented Hilbert space can give rise to superdiffusion. This motivates the study of transport in the particle conserving East model restricted to a particular sector of the Hilbert space.

In this work, we investigate a generalized East model, consisting of hard-core bosons with constrained hopping. The constraint prevents hopping in the absence of bosons on a few preceding sites *to the left*. The chiral nature of such facilitated hopping strongly breaks inversion symmetry, akin to the conventional East model, additionally featuring the conservation of the total number of bosons. Our results show that combining charge conservation and the breaking of inversion symmetry yields new interesting transport phenomena. Specifically, we characterize the proposed generalized East model using its eigenstate properties and dynamics. The detailed study of the eigenstates reveals so-called quantum Hilbert space fragmentation, so far reported only in a few other models [24, 37]. The quantum fragmentation we observe in our model leads to the existence of eigenstates that have zero entanglement along one or several bipartite cuts. The number of these low entanglement eigenstates increases exponentially with system size. We find that these unusual eigenstates can be constructed recursively, relying on special eigenstates existing in small chains that are determined analytically.

The study of dynamics of the particle-conserving East model reveals that weakly entangled eigenstates existing in the spectrum can be probed by quenches from simple product states. In addition, the dynamics from a domain wall initial state reveals two distinct transport regimes. At short times dynamics is superdiffusive, whereas at longer times the constraint leads to a logarithmically slow spreading. We recover the logarithmically slow dynamics within a classically simulable cellular automaton that has the same features as the Hamiltonian model. In contrast, the early time dynamical exponent differs between the quantum Hamiltonian dynamics and the cellular automaton. Additionally, the transport properties show signatures of dependence on the density of particles in the leftmost part of the chain of the initial state. These unusual results call for a more detailed exploration and better understanding of the reported superdiffusive dynamics and its stability in the thermodynamic limit. This invites the systematic study of such models using large scale numerical methods and development of a hydrodynamic description of transport in such systems.

The remainder of the paper is organized as follows. In Section 2 we introduce the Hamiltonian of the particle-conserving East model and explain the effect of the constraint. We then investigate the nature of the Hilbert space fragmentation and of the eigenstates in Section 3. In Section 4 we investigate the dynamical properties of the system, showing similarities in the long-time behavior among the quantum dynamics and the classical cellular automaton. Finally, in Section 5, we conclude by presenting a summary of our work and proposing possible future directions.

## 2 Family of particle-conserving East models

We introduce a family of particle conserving Hamiltonians inspired by the kinetically constrained East model in one dimension. The East model, studied both in the classical [1, 34] and quantum [14, 31, 33] cases, features a constraint that strongly violates inversion symmetry: a given spin is able to flip only if its *left* neighbor is in the up (↑) state. A natural implementation of such a constrained kinematic term in the particle-conserving case is a hopping process *facilitated* by the presence of other particles on the left. The simplest example of such a model is provided by the following Hamiltonian operating on a chain of hard-core bosons,

$$\hat{H}_{r=1} = \sum_{i=2}^{L-1} \hat{n}_{i-1}\left(\hat{c}_i^\dagger \hat{c}_{i+1} + \hat{c}_{i+1}^\dagger \hat{c}_i\right), \tag{1}$$

where the operator $\hat{n}_i = \hat{c}_i^\dagger \hat{c}_i$ is the projector onto the occupied state of site $i$. We assume open boundary conditions here and throughout this work, and typically initialize, without loss of generality, the first site as being occupied by a frozen particle. All sites to the left of the

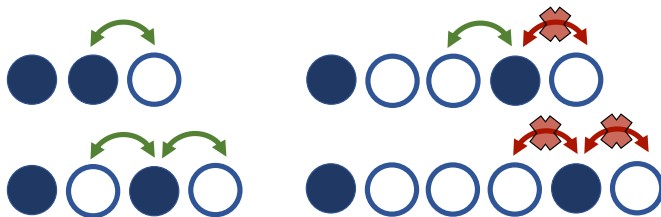

Figure 1: Illustration of constrained hopping in the range-2 particle conserving East model.

leftmost particle, in fact, cannot be occupied, hence they are not relevant to the behavior of the system.

The Hamiltonian (1) implements hopping facilitated by the *nearest neighbor* particle on the left, hence we refer to it as the range-1, $r = 1$, particle conserving East model. A natural extension of this model would be hopping facilitated by the nearest *or* next nearest neighbor, which reads:

$$\hat{H}_2 = \sum_{i=2}^{L-1} (\hat{n}_{i-2} + \hat{n}_{i-1} - \hat{n}_{i-2}\hat{n}_{i-1})(\hat{c}_i^\dagger \hat{c}_{i+1} + \text{H.c.}) , \tag{2}$$

where we treat the operator $\hat{n}_{i=0} = 0$ as being identically zero. Note, that in this Hamiltonian we use the same hopping strength irrespective if the facilitating particle is located on the nearest neighbor or next nearest neighbor site, however this condition may be relaxed. Examples of range-1, $\hat{H}_1$, and range-2, $\hat{H}_2$, particle conserving East models can be further generalized to arbitrary range $r$ as

$$\hat{H}_r = \sum_{i=r+1}^{L-1} \hat{\mathcal{K}}_{i,r}(\hat{c}_{i+1}^\dagger \hat{c}_i + \text{H.c.}), \tag{3}$$

$$\hat{\mathcal{K}}_{i,r} = \sum_{\ell=1}^{r} t_\ell \hat{\mathcal{P}}_{i,\ell} , \tag{4}$$

where the operator $\hat{\mathcal{K}}_{i,r}$ implements a range-$r$ constraint using projectors on the configurations with $\hat{n}_{i-\ell} = 1$ and the region $[i-\ell+1, i-1]$ empty, $\hat{\mathcal{P}}_{i,\ell} = \hat{n}_{i-\ell} \prod_{j=i-\ell+1}^{i-1}(1-\hat{n}_j)$. The coefficients $t_\ell$ correspond to the amplitude of the hopping facilitated by the particle located $\ell$-sites on the left. The Hamiltonian $\hat{H}_2$ in Eq. (2) corresponds to the particular case when all $t_\ell = 1$.

Models with similar facilitated hopping terms were considered in the literature earlier. In particular a pair hopping $\bullet\bullet\circ \leftrightarrow \circ\bullet\bullet$ was introduced in [38] and later used in [39] to probe many-body mobility edges, and shown to be integrable in Ref. [25]. In [40] a similar constrained hopping term was shown to arise from the Jordan-Wigner transformation of a next nearest neighbor XY spin chain. Another constrained model recently studied is the so-called *folded* XXZ [16, 21], where the $\Delta \to \infty$ limit of the XXZ chain is considered, leading to integrable dynamics [22, 23]. The key difference in our work, compared to the previous literature, consists of having a chiral kinetic term, whereas in the mentioned works symmetric constraints are considered.

Hamiltonians $\hat{H}_r$ for all values of $r$ feature $U(1)$ symmetry related to the conservation of total boson number, justifying the name of particle-conserving East models. In this work we mostly focus on the case of $r = 2$ with homogeneous hopping parameters $t_\ell = 1$, as written in Eq. (2). We discuss the generality of our results with respect to the choice of hopping strengths and range of constraint in Appendices D and F. A major feature of this family of models is Hilbert space fragmentation, which is known to affect spectral and dynamical properties. As such we begin our investigation by looking into the nature of Hilbert space fragmentation in

these models in Section 3, where we highlight the generality of our results, by formulating them for a general range $r$ and show examples for $r = 2$.

# 3 Hilbert space fragmentation and eigenstates

In this Section we focus on the phenomenon of Hilbert space fragmentation in the particle-conserving East models introduced above. First, we discuss the block structure of the Hamiltonian in the product state basis — known as classical Hilbert space fragmentation — and define the largest connected component of the Hilbert space. Next, in Sec. 3.2 we discuss the emergence of disconnected components of the Hilbert space that are not manifest in the product state basis, leading to quantum Hilbert space fragmentation.

## 3.1 Classical Hilbert space fragmentation

Due to the $U(1)$ symmetry of the Hamiltonian (3), the global Hilbert space is divided in blocks labeled by the different number of bosons $N_p$ with dimension given by the binomial coefficient $\mathcal{C}_{N_p}^L$. Within each given sector of total particle number $N_p$, the constrained hopping causes further fragmentation of the Hilbert space in extensively many subspaces. First, the leftmost boson in the system is always frozen. Hence, as we discussed in Section 2, we choose the first site to be always occupied, which may be viewed as a boundary condition. In addition, a boson may also be frozen if the number of particles to its left is too small. An example configuration is given by the product state $| \bullet \circ \circ \circ \bullet \bullet \circ \circ \rangle$ for the $r = 2$ model, where $\circ$ corresponds to an empty site and $\bullet$ is a site occupied by one boson. Here the second boson cannot move since the previous two sites are empty and cannot be occupied.

In view of this additional fragmentation, we focus on the largest classically connected sector of the Hilbert space with a fixed number of particles, $N_p$. This sector can be constructed starting from a particular initial state $|\text{DW}\rangle$, where all particles are located at the left boundary,

$$|\text{DW}\rangle = |\underbrace{\bullet \bullet \bullet \cdots \bullet}_{N_p} \underbrace{\circ \circ \circ \cdots \circ}_{L-N_p}\rangle . \tag{5}$$

Starting from this initial state the constraint will limit the spreading of particles, that can reach at most

$$L_r^*(N_p) = (r+1)N_p - r \tag{6}$$

sites, corresponding to the most diluted state, $| \bullet \circ \circ \bullet \circ \circ \bullet \circ \circ \bullet \ldots \rangle$ for $r = 2$. Thus, in what follows we use the system size $L = L_r^*$ uniquely defined by the number of particles and the range of the constraint in Eq. (6).

The fragmentation of the Hilbert space discussed above may be attributed to a set of emergent conserved quantities in the model in addition to the total particle number, $\hat{N}_{\text{tot}} = \sum_i \hat{n}_i$. The first class of conserved operators responsible for the freezing of the leftmost particle is written as

$$\hat{N}_{\ell_0} = \ell_0 \left[ \prod_{i < \ell_0} (1 - \hat{n}_i) \right] \hat{n}_{\ell_0} . \tag{7}$$

Since projectors in this operator are complementary to the projectors in the Hamiltonian, this satisfies the property $\hat{N}_{\ell_0} \hat{H}_r = \hat{H}_r \hat{N}_{\ell_0} = 0$, hence trivially having a zero commutator. This conservation law induces further fragmentation of the Hilbert space into $L - N_p$ sectors labeled by the position of the leftmost boson.

The second class of operators yields a further fragmentation within each sector with a fixed position of the leftmost particle. Bearing in mind that the leftmost compact cluster of

$\tilde{N}$ particles cannot expand farther than $\tilde{L} = L_r^*(\tilde{N})$, one can realize that if $r+1$ sites or more are left empty to the right of $\tilde{L}$ then the chain is dynamically separated into two independent regions. The $\tilde{N}$ particles on the left cannot spread to the right side $i \geq \tilde{L} + r$ as well as the leftmost particle on the right cannot move to the left as the constraint is never fulfilled. The simplest example of such a configuration is given by $|\bullet \circ \circ \bullet \circ \circ \circ \bullet \dots\rangle$ for $r = 2$, $\tilde{N} = 2$ and $\tilde{L} = 4$. Crucially, the position $\tilde{j} > \tilde{L} + r$ of the first occupied site on the right can be chosen arbitrarily, as long as it satisfies the global constraints of the system. Formally, then, one can define a family of conserved quantities given by the projector $\hat{\mathcal{P}}_{\tilde{N}}$ on configurations with $\tilde{N}$ particles in the leftmost $\tilde{L}$ sites followed by a sufficiently large empty region, and, finally, an occupied site $\tilde{j}$

$$\hat{O}_{\tilde{N}}^{\tilde{j}} = \hat{\mathcal{P}}_{\tilde{N}, \tilde{i}} \left[ \prod_{k=\tilde{L}+1}^{\tilde{j}-1} (1 - \hat{n}_k) \right] \hat{n}_{\tilde{j}}. \tag{8}$$

The freedom in the choice of $\tilde{j}$ yields $r(N_p - \tilde{N} - 1)$ different sectors for a fixed $\tilde{N}$. Hence, the number of fragmented sectors is given by

$$\sum_{\tilde{N}=1}^{N_p-1} r(N_p - \tilde{N} - 1) = \left[ \frac{1}{2}(N_p^2 - 3N_p) + 1 \right] \propto N_p^2. \tag{9}$$

We notice that additional levels of fragmentation can emerge whenever the right part can be further decomposed in a similar way to the one discussed above. This corresponds to composing two different $\hat{O}_{\tilde{N}}^{\tilde{j}}$ where the second is shifted by $\tilde{j}$ sites. Every time that happens, additional subsectors appear for some of the sectors identified by the operator $\hat{O}_{\tilde{N}}^{\tilde{j}}$. As the number of additional levels of fragmentation increases proportionally to $N_p$, each adding subsectors to the previous level, one finally obtains that the asymptotic behavior of the global number of classically fragmented subsectors has to be $O(\exp(N_p))$. The exponential increase of the number of disconnected subsectors was verified numerically, thus properly identifying a case of Hilbert space fragmentation. Finally, we note that in our case, the operators defined in Eq. (8) do not commute with each of the individual terms of the Hamiltonian, as in the definition of Ref. [37]. Nevertheless, they define an algebra of conserved quantities whose size grows exponentially with system size.

## 3.2 Quantum Hilbert space fragmentation

Due to the fragmentation of the Hilbert space in the computational basis discussed above, we focus on the largest sector of the Hilbert space as defined in the previous section. In Appendix A we show that the statistic of the level spacing for the Hamiltonian $\hat{H}_2$ within this block follows the Wigner-Dyson surmise, confirming that we resolved all symmetries of this model and naïvely suggesting an overall thermalizing (chaotic) character of eigenstates [41].

To further check the character of eigenstates, we consider their entanglement entropy. We divide the system into two parts, $A$ containing sites $1, \dots, i$, $A = [1, i]$ and its complement denoted as $B = [i+1, L]$. The entanglement entropy of the eigenstate $|E_\alpha\rangle$ for such bipartition is obtained as the von Neumann entropy of the reduced density matrix $\rho_i = \mathrm{tr}_B |E_\alpha\rangle\langle E_\alpha|$

$$S_i = -\mathrm{tr}\left[\rho_i \ln \rho_i\right]. \tag{10}$$

In thermal systems entanglement of highly excited eigenstates is expected to follow volume law scaling, increasing linearly with $i$ for $i \ll L$, and reaching maximal value for $i = L/2$. However, our numerical study of the entanglement entropy shows strong deviations from these expectations, in particular revealing a significant number of eigenstates with extremely low,

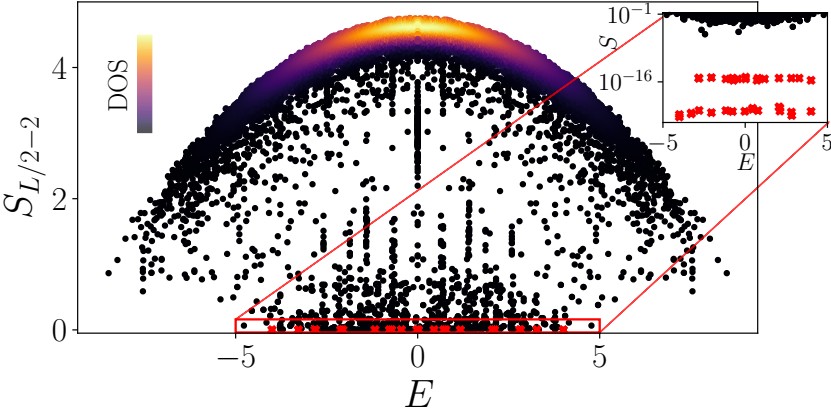

Figure 2: Entanglement entropy of eigenstates along the bipartite cut at the site 8 for $N_p = 8$ and $L = 22$. The color intensity corresponds to the density of dots, revealing that the majority of the eigenstates have nearly thermal entanglement. However, a large number of eigenstates has entanglement much lower than the thermal value. Among these, the red crosses correspond to entanglement being zero up to numerical precision (inset).

and even exactly zero, entanglement, a feature typical of quantum many-body scars [26–28, 42–48].

Figure 2 illustrates such anomalous behavior of eigenstate entanglement for a chain of $L = 22$ sites. For the bipartite cut shown, $A = [1, 8]$, most of the eigenstates have increasing entanglement as their energy approaches zero, where the density of states is maximal, in agreement with thermalization. Nevertheless, a significant number of eigenstates features much lower values of entanglement, and the red box and inset in Fig. 2 highlight the presence of eigenstates with zero entanglement (up to numerical precision). We explain this as a result of an additional fragmentation of the Hilbert space caused by the interplay of the constraint and boson number conservation.

Eigenstates with zero entanglement, denoted as $|E_{S=0}\rangle$, are separable and can be written as a product state of the wave function in the region $A$ and in its complement $B$. To this end, we choose the wave function $|\psi_m^\ell\rangle$ of the separable state $|E_{S=0}\rangle$ in the region $A$ as an eigenstate of the Hamiltonian $\hat{H}_r$ restricted to the Hilbert space of $m$ particles in $\ell$ sites. The state $|\psi_m^\ell\rangle$ has to satisfy the additional condition $\langle\psi_m^\ell|\hat{n}_\ell|\psi_m^\ell\rangle = 0$, i.e. that the last site of the region is empty. Provided such state exists, we construct the separable eigenstate $|E_{S=0}\rangle$ as

$$|E_{S=0}\rangle = |\psi_m^\ell\rangle \otimes \underbrace{|\circ\,\circ\cdots\circ\rangle}_{q} \otimes |\psi_R\rangle \,, \tag{11}$$

where $|\psi_R\rangle$ is an eigenstate of the Hamiltonian restricted to $L-\ell-q$ sites and $N_p-m$ particles. Inserting an empty region of $q \geq r$ sites separating the support of $|\psi_m^\ell\rangle$ and $|\psi_R\rangle$ ensures that the two states are disconnected. Note that $q$ is upper bounded by the requirement that the resulting state belongs to the largest classically fragmented sector. It is easy to check that the state $|E_{S=0}\rangle$ is an eigenstate of the full Hamiltonian. Indeed, thanks to the empty region $q$ the particles in $A$ cannot influence those in $B$ and the two eigenstates of the restricted Hamiltonian combine into an eigenstate of the full system.

Similarly to the case of classical fragmentation discussed in Eq. (8), one can define a family of operators that commute with the Hamiltonian and label the different sectors arising due to

quantum fragmentation

$$\hat{O}^q_{\psi^\ell_m} = \hat{\mathcal{P}}_{\psi^\ell_m} \left[ \prod_{k=\ell+1}^{\ell+q+1} (1 - \hat{n}_k) \right] \hat{n}_{\ell+q+2}, \tag{12}$$

where $\hat{\mathcal{P}}_{\psi^\ell_m}$ is the projector onto the eigenstate of the restricted Hamiltonian $|\psi^\ell_m\rangle$.

The construction of $|E_{S=0}\rangle$ relies on the existence of eigenstates $|\psi^\ell_m\rangle$ with vanishing density on the last site. This is a non-trivial requirement that *a priori* is not expected to be satisfied. However, we observe that such eigenstates can be found within the degenerate subspace of eigenstates with zero energy, see Appendix C. If $|\psi^\ell_m\rangle$ is an eigenstate with zero energy, the energy of eigenstate $|E_{S=0}\rangle$ is determined only by the energy of the $|\psi_R\rangle$. The existence of $|\psi^\ell_m\rangle$ relies on two conditions which have to hold simultaneously: $\ell > m + r$ and $(r+1)m - r \geq \ell$. These are satisfied only for $m \geq 3$ particles, thus resulting in a minimal size of the left region $\ell_{\min} = 6$ for $r = 2$. While there is no guarantee that states $|\psi^\ell_m\rangle$ exist for generic $(m, \ell)$, we have an explicit analytic construction for the smallest state $|\psi^6_3\rangle$ for $(m, \ell) = (3, 6)$

$$|\psi^6_3\rangle = \frac{1}{\sqrt{2}} \left[ | \bullet\bullet\circ\circ\bullet\circ\rangle - | \bullet\circ\bullet\bullet\circ\circ\rangle \right], \tag{13}$$

similarly we report solutions up to $(m, \ell) = (7, 18)$ in Appendix C. Furthermore, for each $(m, \ell)$ satisfying the condition, one can easily verify that stacking multiple $|\psi^\ell_m\rangle$ separated by at least $r$ empty sites generates another state fulfilling the same condition. This recursive construction of the left states in Eq. (11), together with the explicit example Eq. (13), guarantees the existence of an infinite number of $|\psi^\ell_m\rangle$, in the thermodynamic limit. We further notice that a similar decomposition can be applied to the right eigenstates, $|\psi_R\rangle$ in a recursive fashion.

The construction of the eigenstates described above suggests that combining two operators $\hat{O}_{\psi^\ell_m}$, one shifted by $\ell + q$ sites, yields a new operator commuting with the full Hamiltonian and labeling a different fragmented subsector. Since there exist at least two different $|\psi^\ell_m\rangle$, one can combine them in various ways always obtaining new sectors. Due to this property, the size of the algebra of operators $\hat{O}_{\psi^\ell_m}$ scales as the total number of such combinations, which increases exponentially with system size in the thermodynamic limit. We observe that the operators defined in Eq. (12) do not commute with each individual term of the Hamiltonian, as required by the definition of Ref. [37]. However, they still give rise to a block-diagonal Hamiltonian in the *entangled* basis resulting from the product of the eigenstates of the restricted Hamiltonian on the left with product states on the remainder of the system, thus presenting a genuine case of quantum Hilbert space fragmentation. The recursive nature of the construction of constrained eigenstates might indicate the existence of a more general structure, possibly common to models featuring both particle conservation and chiral constraints. Hence, the formal definition of such *recursive* Hilbert space fragmentation presents an interesting direction for future work.

Let us explore the consequence of the existence of the special eigenstates defined in Eq. (11). Given the special character of the wave function $|\psi^\ell_m\rangle$, we expect that states $|E_{S=0}\rangle$ have a similar pattern of local observables in the first $\ell$ sites. An example of such behavior is shown in Figure 3(a), which reveals that all four states $|E_{S=0}\rangle$ that have zero entanglement across at least one bipartite cut in the $L = 13$ chain for $r = 2$ feature the same density expectation values, $\langle \hat{n}_i \rangle_\alpha = \langle E_\alpha | \hat{n}_i | E_\alpha \rangle$, in the first $\ell = 6$ sites. Starting from the site number $i = 9$, the density profile has different values on different eigenstates, corresponding to different wave functions $|\psi_R\rangle$ in Eq. (11).

The number of eigenstates with zero entanglement grows exponentially with system size. Even for the case of a fixed $|\psi^\ell_m\rangle$, the right restricted eigenstate $|\psi_R\rangle$ is not subject to any additional constraints, hence the number of possible choices of $|\psi_R\rangle$ grows as the dimension

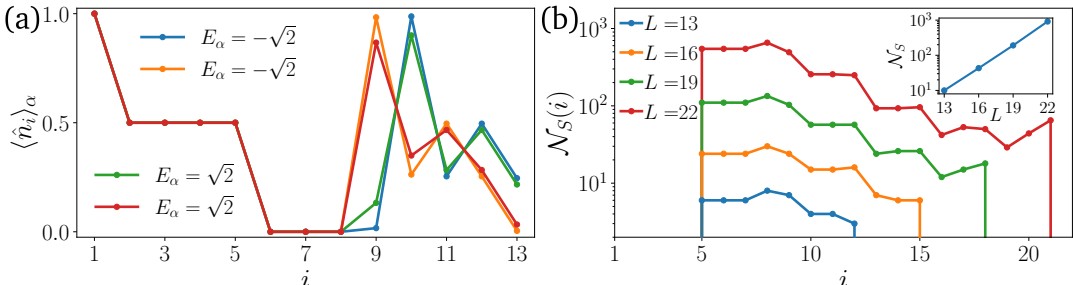

Figure 3: (a): The density profile of the zero-entanglement eigenstates for $L = 13$ shows a common pattern, due to their special structure (11). The first sites correspond to the zero mode of the Hamiltonian restricted to 3 particles in 6 sites $|\psi_3^6\rangle$, followed by 2 empty sites. The right subregion can then be any of the 6 eigenstate of $H$ for 2 particles in 4 sites, with energy $\pm\sqrt{2}$, 0. We note that eigenstates with the same $|\psi_R\rangle$ but a different number of empty sites separating it from $|\psi_m^\ell\rangle$ are degenerate and can be mixed by the numerical eigensolver, as is the case in the density profiles shown here. (b): The number of zero entanglement entropy eigenstates $\mathcal{N}_S(i)$ depends on the boundary of the subregion $A = [1, i]$. In particular, in the interval $i \in [5, 9]$ the number of zero-entanglement eigenstates is exponentially larger compared to more extended left subregions. At larger $i$ recursively fragmented eigenstates contribute to $\mathcal{N}_S(i)$ for $L \geq 13$. The total number of zero-entanglement eigenstates, $\mathcal{N}_S$, grows exponentially in $L$, as shown in the inset. Note that $\mathcal{N}_S \neq \sum_i \mathcal{N}_S(i)$, as some eigenstates have zero entanglement across multiple bipartite cuts.

of the Hilbert space of $N_p - m$ particles on $L - \ell - r$ sites, that is, at fixed $m$, asymptotically exponential in $N_p$. In the general case where $(m, \ell)$ are allowed to change, new $|E_{S=0}\rangle$ states will appear, with zero entanglement entropy at different bipartite cuts, according to the size of the left region. Finally, the recursive nature of the fragmentation discussed above is expected to give eigenstates with zero entropy across two or more distinct cuts which are separated by a non-vanishing entanglement region. These states are observed in numerical simulations starting from $N_p = 7$ and $L = 19$.

To illustrate the counting of eigenstates with zero entropy at a cut separating subregion $A = [1, i]$ from the rest of the system, we denote their number as $\mathcal{N}_S(i)$. For $i < 5$, this number is zero $\mathcal{N}_S(i) = 0$, as explained in the construction of these states. For $i \geq 5$ we observe a large $\mathcal{N}_S(i)$, exponentially increasing with system size. However, at larger $i$, the available configurations that can support states of the form Eq. (11) decrease and $\mathcal{N}_S(i)$ drops and eventually vanishes. As $N_p$ and system size increase, left states $|\psi_m^\ell\rangle$ with a larger support $\ell$ are allowed thus increasing the range of sites where $\mathcal{N}_S(i) > 0$. This is also due to recursive fragmentation which can appear starting from $N_p = 5$ and $L = 13$, leading to eigenstates with zero entanglement across at least one cut. Carefully counting all *distinct* eigenstates $|E_{S=0}\rangle$ we confirm that their total number $\mathcal{N}_S$ grows exponentially with system size in the inset of Fig. 3(b).

## 4 Dynamics

After discussing recursive construction of the quantum Hilbert space fragmentation in the particle-conserving East model, we proceed with the study of the dynamics. First, in Section 4.1 we consider the dynamical signatures of Hilbert space fragmentation. Afterwards, in Section 4.2 we discuss the phenomenology of particle spreading starting from a domain wall

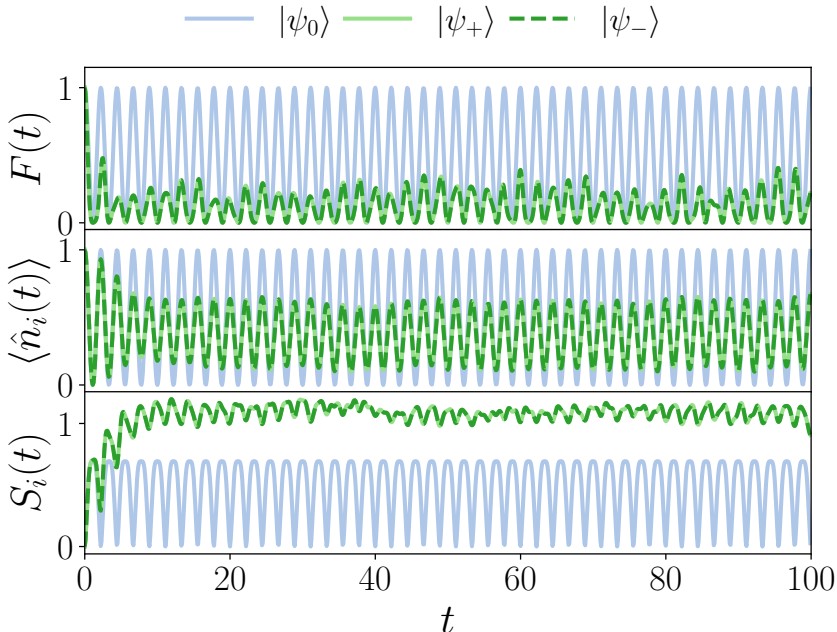

Figure 4: The signatures of quantum Hilbert space fragmentation can be observed for initial states that have a large overlap with zero-entanglement eigenstates $|E_{S=0}\rangle$. The fidelity $F(t) = |\langle\psi_0|\psi(t)\rangle|^2$ shows periodic revivals for all three initial states; choosing an eigenstate on the left portion of the chain results in perfect revivals (blue curve). Entanglement entropy across the cut $i = 11$ in the middle of the right region $R$ and density on the same site show oscillations with identical frequency.

state and illustrate how this can be connected to the structure of the Hilbert space. Finally, we compare the quantum dynamics to that of a classical cellular automaton in Section 4.3.

## 4.1 Dynamical signatures of quantum Hilbert space fragmentation

The zero-entanglement eigenstates $|E_{S=0}\rangle$ identified in Eq. (11) span a subsector of the Hilbert space which is dynamically disconnected from the rest. In this subspace the Hamiltonian has non-trivial action only in the right component of the state, and eigenstates can be written as product states across the particular cut. Below we discuss signatures of such fragmentation in dynamics launched from weakly entangled initial states.

As an illustrative example, we show in Figure 4 the time evolution of a state of the form defined in Eq. (11) for $L = 13$. To obtain non-trivial dynamics, we replace the eigenstate $|\psi_R\rangle$ with a product state. In particular, we choose the initial state as

$$|\psi_0\rangle = \frac{|\bullet\bullet\circ\circ\bullet\circ\rangle - |\bullet\circ\bullet\bullet\circ\circ\rangle}{\sqrt{2}} \otimes |\circ\circ\rangle \otimes |\bullet\circ\bullet\circ\circ\rangle, \tag{14}$$

and consider the time-evolved state $|\psi(t)\rangle = e^{-\imath t \hat{H}_2}|\psi_0\rangle$. The action of the full Hamiltonian does not affect the left part of the state and the Hamiltonian acting on the last five sites in the chain $R = [9, 13]$ is a simple $3 \times 3$ matrix

$$\hat{H}_R = \begin{pmatrix} 0 & 1 & 0 \\ 1 & 0 & 1 \\ 0 & 1 & 0 \end{pmatrix}, \tag{15}$$

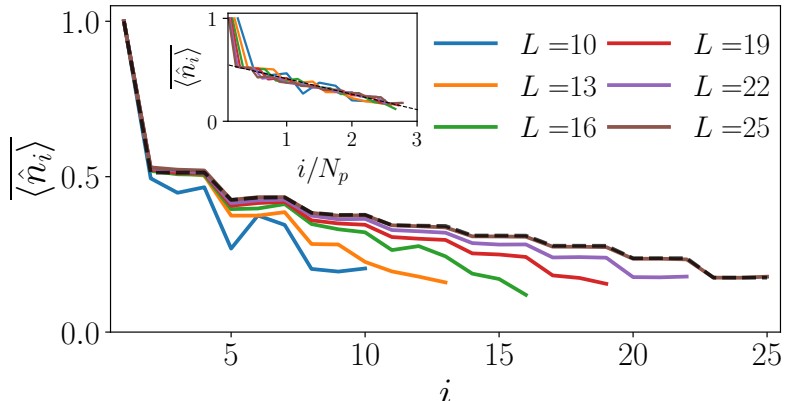

Figure 5: The constrained character of the model leads to a non-uniform stationary density profile for the domain wall initial state. This coincides with the infinite-temperature prediction on large systems, as highlighted by the dashed line corresponding to $\mathrm{tr}[\hat{n}_i]/\mathrm{tr}[\mathbb{1}]$ for $L = 25$, where $\mathrm{tr}[\hat{O}] = \sum_j \hat{O}_{jj}$. Rescaling the $x$-axis by the number of particles $N_p$, we obtain a good collapse of the data, as shown in the inset. The particle density follows a linear decrease $\overline{\langle \hat{n}_i \rangle} \approx \overline{\langle \hat{n}_2 \rangle} - c(i-2)/N_p$, with $c \approx 0.15$.

in the $\{|\bullet\bullet\circ\circ\circ\rangle, |\bullet\circ\bullet\circ\circ\rangle, |\bullet\circ\circ\bullet\circ\rangle\}$ basis. Diagonalizing this matrix, we write the time-evolved state $|\psi_0(t)\rangle$ as

$$|\psi(t)\rangle = |\psi_m^\ell\rangle \otimes |00\rangle \otimes \left[ \cos(\sqrt{2}t)|\bullet\circ\bullet\circ\circ\rangle - \sin(\sqrt{2}t)\frac{|\bullet\bullet\circ\circ\circ\rangle + |\bullet\circ\circ\bullet\circ\rangle}{\sqrt{2}} \right], \qquad (16)$$

hence the fidelity reads $F(t) = |\langle\psi_0|\psi(t)\rangle|^2 = \cos^2(\sqrt{2}t)$. As the time-evolution in Eq. (16) involves only three different product states, it produces perfect revivals with period $T = \pi/\sqrt{2}$. This periodicity also affects observables, such as the density in the region $R$, and the entanglement entropy.

This periodic dynamics also appears in the two product states $|\psi_+\rangle = |\bullet\bullet\circ\circ\bullet\circ\circ\circ\bullet\circ\bullet\circ\circ\rangle$ and $|\psi_-\rangle = |\bullet\circ\circ\bullet\bullet\circ\circ\circ\circ\bullet\circ\bullet\circ\circ\rangle$ that are contained in Eq. (14). These states indeed show revivals of the fidelity with the same period $T$, although the peaks are more suppressed. This is not surprising, as these states have only part of their weight in the disconnected subspace.

In Figure 4 we show the results of the dynamics of the state $|\psi_0\rangle$, Eq. (14), together with the two product states generating the superposition, $|\psi_\pm\rangle$. In addition to fidelity, we also show the density and entanglement dynamics of sites $i$ within the right region $R$. As expected, the fidelity shows revivals with period $T = \pi/\sqrt{2}$, and similar oscillations are also observed in local operators and entanglement. While the initial state $|\psi_0\rangle$ defined in Eq. (14) presents perfect revivals with $F(T) = 1$, the product states $|\psi_\pm\rangle$ do not display perfect fidelity revivals and show larger entanglement. We note, that since the two product states $|\psi_\pm\rangle$ together form a state $|\psi_m^\ell\rangle$ their dynamics in the region $R$ is not affected by the choice of the left configuration, and all considered quantities for theses two initial states have identical dynamics.

## 4.2 Phenomenology of dynamics from the $|\mathrm{DW}\rangle$ initial state

After exploring the dynamics resulting from quantum Hilbert space fragmentation, we now turn to the dynamics in the remainder of the constrained Hilbert space focusing on the domain wall state (5). The domain wall state does not have any overlap with zero entanglement eigenstates except for possibly states with zero entanglement on the last cut. It is also characterized by a vanishing expectation value of the Hamiltonian, corresponding to zero energy

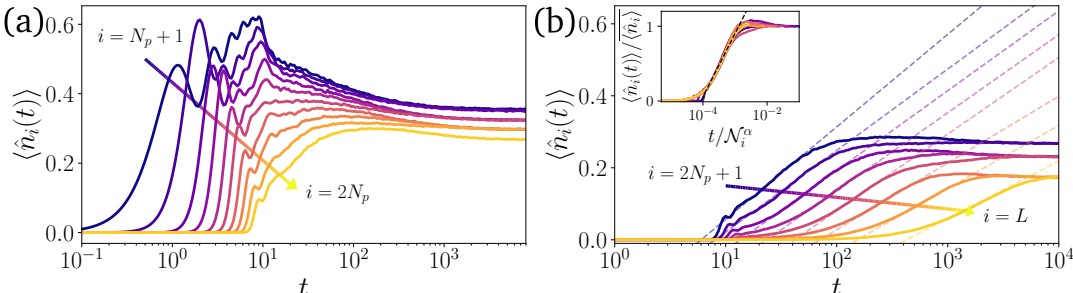

Figure 6: The approach to saturation in the density dynamics is very different depending on the region within the chain. (a) In the first $2N_p$ sites of the chain a fast relaxation takes place due to the weak role of the constraint in dense regions. (b) For the right part of the chain, $i > 2N_p$ anomalously slow logarithmic dynamics arise. The inset shows the data collapse upon rescaling the density axis by the long time average and the time axis by the number of states within each *leg* of the graph $\mathcal{N}_i$ to the power $\alpha \approx 1.15$, as discussed in more detail at the end of this section. The data shown here are for a system of $L = 28$ sites with $N_p = 10$ bosons.

density, where the density of states is maximal. Hence, thermalization implies that time evolution from the domain wall leads to the steady state where all observables agree with their infinite-temperature expectation value. To check this property we focus on the expectation value of the particle density operators throughout the chain.

Figure 5 shows the infinite time average of the particle density, $\overline{\langle \hat{n}_i \rangle}$ obtained through the diagonal ensemble

$$\overline{\langle \hat{n}_i \rangle} = \sum_{\alpha} |\langle \mathrm{DW} | E_\alpha \rangle|^2 \langle E_\alpha | \hat{n}_i | E_\alpha \rangle \,, \tag{17}$$

where the sum runs over all eigenstates $\alpha$. This calculation is performed for $L \leq 22$, where the full set of eigenstates can be obtained through exact diagonalization. For larger systems, the infinite time average value of $\overline{\langle \hat{n}_i \rangle}$ is approximated as the average of the density in the time-window $t \in [6.9 \times 10^3, 10^4]$. We observe that the density profile agrees well with the infinite-temperature prediction. See Appendix A for details of the calculation.

The infinite-temperature prediction for the density profile does not result in a homogeneous density due to the constraint. The number of allowed configurations with non-zero density in the last sites is indeed limited by the constraint, and results in a lower density in the rightmost parts of the chain. In addition, the profile has a step-like shape that is related to the range-2 constraint in the model. In the inset of Fig. 5 we show that the density profiles collapse onto each other when plotted as a function of $i/N_p$. This suggests the heuristic expression for the density profile $\overline{\langle \hat{n}_i \rangle} \approx \overline{\langle \hat{n}_2 \rangle} - c(i-2)/N_p$ where $c \approx 0.15$ is a positive constant.

Although the saturation profile of the density is consistent with thermalization, below we demonstrate that *relaxation* to the steady state density profile is anomalous. The time-evolution of the density $\langle \hat{n}_i(t) \rangle = \langle \psi(t) | \hat{n}_i | \psi(t) \rangle$ is shown in Figure 6 for $L = 28$ sites up to times $t \approx 10^4$. The data demonstrates that the relaxation of density qualitatively depends on the location within the chain. In the left part of the chain with $i \leq 2N_p$, the spreading of the density front is fast, and saturation is reached quickly on timescales of $O(10)$, as shown in Fig. 6(a). This can be attributed to the fact that the constraint is not effective at large densities. In contrast, in the rightmost part of the chain, $i > 2N_p$ the constraint dramatically affects the spreading of particles resulting in the logarithmically slow dynamics in Fig. 6(b).

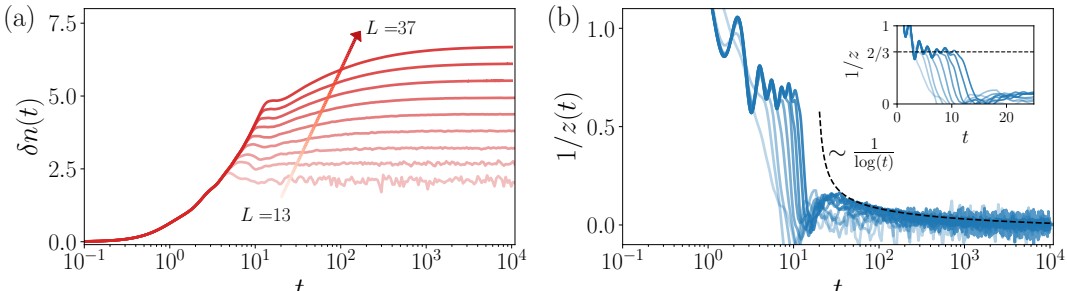

Figure 7: (a) The behavior of the particle current across the domain wall shows an initial power-law growth $\delta n(t) \sim t^{1/z(t)}$ followed by a slow-down to logarithmic behavior at later times, in agreement with the density dynamics. (b) The analysis of the dynamical exponent $z(t)$ shows the presence of a super-diffusive plateau $1/z \approx 2/3$ at intermediate times, whose duration grows linearly with system size. At later times, the onset of logarithmic dynamics is signalled by the decay of $1/z(t)$. Data are for $13 \le L \le 37$ from more to less transparent.

To further characterize the anomalous dynamics, we study the transport of the particle density on short time-scales for larger systems up to $L = 37$ sites. For the systems with $L > 28$ we use a fourth-order Runge-Kutta algorithm with a time-step as small as $\delta t = 10^{-3}$. This allows us to reliably study the short-time behavior with sufficient accuracy down to $\delta t^4 = 10^{-12}$. We consider the particle flow across the domain wall

$$\delta n(t) = \sum_{i \le N_p} \left[ \langle \hat{n}_i(0) \rangle - \langle \hat{n}_i(t) \rangle \right]. \tag{18}$$

The dynamics of $\delta n(t)$ in Figure 7(a) shows a clear initial power-law behavior drifting to much slower logarithmic growth at later times, in agreement with the dynamics of $\langle \hat{n}_i(t) \rangle$ in the right part of the chain. At even longer times $\delta n(t)$ saturates to a value proportional to the system size $L$. Figure 7(b) shows the instantaneous dynamical exponent,

$$z(t) = \left( \frac{d \ln \delta n(t)}{d \ln t} \right)^{-1}. \tag{19}$$

In this figure, the early time dynamics are characterized by fast transport of particles across the domain wall $i = N_p$ due to the large initial density. On intermediate time-scales $t \approx 10$, a superdiffusive plateau of $1/z(t) \approx 2/3$ is visible. Finally, at longer times the dynamics slow down and become logarithmic, consistent with a vanishing $1/z(t)$. Zooming in the time-window $t \le 30$, we notice that the extent of the superdiffusive plateau increases linearly with system size, suggesting the persistence of the superdiffusive regime in the thermodynamic limit.

The superdiffusive behavior observed in the dynamics of the domain wall initial state is very peculiar, as one would expect a state close to infinite temperature to show diffusive transport in an ergodic system as is the one considered here. We further investigated the time-evolution of other initial states with a lower density of particles in the leftmost part of the chain. Appendix E shows that with decreasing density, the transport is drifting from superdiffusion in dense states to diffusion as the density decreases. Such dependence of transport on the density in the initial state suggests that superdiffusive dynamics is related to the special nature of the domain wall state that separates completely empty and full regions, and thus it may lack a coarse-grained hydrodynamics description.

We now focus on capturing the phenomenology of the slow dynamics observed at late times using the structure of the Hamiltonian. Starting from the domain wall initial state,

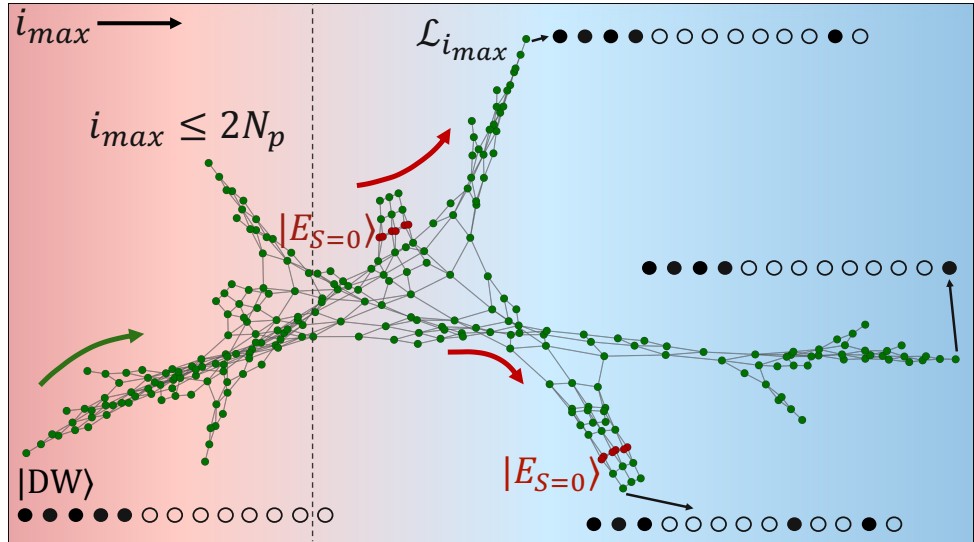

Figure 8: The representation of $\hat{H}_2$ as the adjacency graph $\mathcal{G}_r$ for system with $N_p = 5$ particle and $L = 13$ lattice sites. The dense central part – *backbone* – has gradually decreasing number of vertices and connectivity as the position of the rightmost particle increases above $i_{\max} > 2N_p = 10$ (dashed line). The *legs* of the graph emanate from the backbone and correspond to regions where $i_{\max}$ is conserved. The legs end with the product states (an example is labeled as $\mathcal{L}_{i_{\max}}$), where a particular particle is frozen near the end of the chain. Red vertices show product states corresponding to zero-entanglement eigenstates $|E_{S=0}\rangle$, which in this case have weight on 12 out of $\mathcal{D}_{N_p} = 273$ product states contained in the constrained Hilbert space.

the slow dynamical regime is reached after a time scaling proportionally to the system size. Naively this may preclude the observation of such dynamical regime in the thermodynamic limit. However, our study of transport in more dilute initial states suggests that the onset of logarithmic slow dynamics may depend on the density of particles. In particular, we conjecture that at sufficiently low initial densities, logarithmic dynamics may be observable at timescales that do not depend on the system size. In order to construct a phenomenological picture of slow dynamics, we interpret the Hamiltonian as a graph where the vertices of the graph enumerate the product states contained in a given connected sector of the Hilbert space. The edges of the graph connect product states that are related by any particle hopping process allowed by the constraint. A particular example of such a graph for the system with $N_p = 5$ particles and $L = 13$ sites is shown in Fig. 8.

The vertices of the graph in Fig. 8 are approximately ordered by the position of the rightmost occupied site $i_{\max} \geq N_p$, revealing the particular structure emergent due to the constraint. The dense region that follows the domain wall product state has high connectivity, and we refer to it as the *backbone*. In addition to the backbone, the graph has prominent *legs* emanating perpendicularly. The legs are characterized by the conserved position of the rightmost particle that is effectively frozen due to the particles on the left retracting away, as pictorially shown in Fig. 8. Since such legs are in one-to-one correspondence with the position of the rightmost particle, $i_{\max}$, their number grows linearly with system size. The number of product state configurations contained within each leg strongly depends on $i_{\max}$. Given that the position of the rightmost particle is frozen within a leg, they cast a strong effect on the dynamics of the model.

In particular, the spreading of particles towards the right probed by $R(t)$ can be related to the presence of an increasing number of configurations within legs at large $i_{\max}$, $\mathcal{N}_{i_{\max}}$. These are characterized by long empty regions as the one depicted in Figure 8, which require

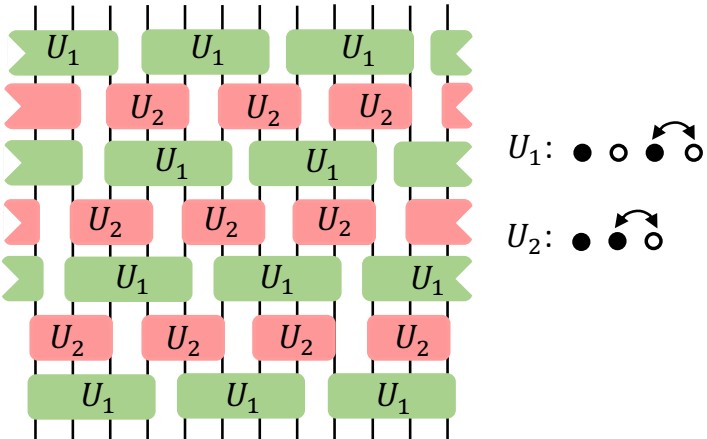

Figure 9: Schematic representation of the circuit used to describe the classical dynamics. The continuous time-evolution $\hat{U}(t)$ is decomposed into a series of 4-sites gates $U_1$ and of 3-sites gates $U_2$, whose action is shown on the right part of the Figure.

the collective motion of many particles to allow the hopping of the rightmost boson sitting at $i_{\max}$. The slow dynamics observed, then, can be qualitatively understood as the effect of many states not contributing to the spreading and of the increasingly long empty regions that have to be crossed to activate hopping further to the right. Looking back at the dynamics shown in Figure 6, we highlight this effect by rescaling the time-axis by the number of configurations belonging to each leg, $\mathcal{N}_i$. The resulting collapse is shown in the inset of Figure 6(b).

## 4.3 Dynamics in constrained classical cellular automata

The anomalous relaxation of the quantum model from the domain wall state reported in Section 4.2 invites natural questions about the universality of dynamics in presence of inversion-breaking constraints. To shed light on this question, we introduce a classical cellular automaton model that replaces the unitary time-evolution of the quantum model $\hat{U}(t) = \exp(-\iota \hat{H} t)$ with a circuit of local unitary gates preserving the same symmetries and constraints of the Hamiltonian [49, 50].

To reproduce correlated hopping in the Hamiltonian (2), we introduce two sets of local gates $U_1$ and $U_2$ schematically shown in Fig. 10(a). The first gate, $U_1$, acts on 4 sites and implements the hopping facilitated by the next nearest neighbor,

$$U_1 = \exp\left\{ -\iota\theta\left[ \hat{n}_j(1-\hat{n}_{j+1})\big(c^\dagger_{j+3}c_{j+2} + \text{H.c.}\big)\right]\right\}. \tag{20}$$

The second gate, $U_2$, acts on three sites, and implements the hopping facilitated by the nearest neighbor site:

$$U_2 = \exp\left\{ -\iota\theta\left[ \hat{n}_j\big(c^\dagger_{j+2}c_{j+1} + \text{H.c.}\big)\right]\right\}. \tag{21}$$

For a generic choice of the rotation angle $\theta$ these gates cannot be efficiently simulated classically. However, in what follows we fix $\theta$ to the special value, $\theta = \pi/2$, so that gates $U_{1,2}$ map any product state to another product state. This corresponds to a classical cellular automaton which allows for efficient classical simulation.

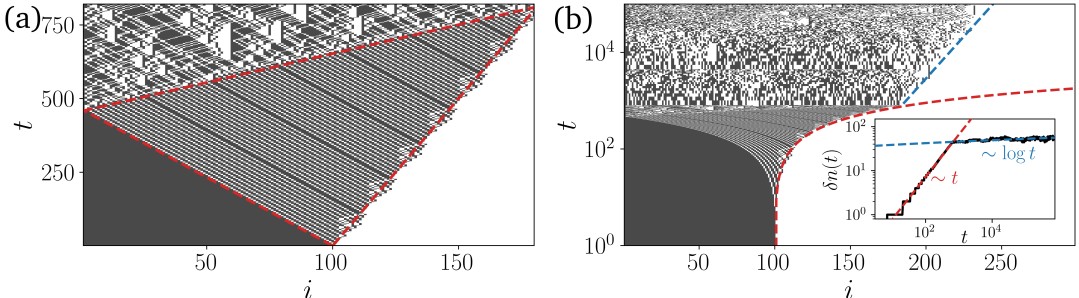

Figure 10: (a)-(b): Density evolution of the classical cellular automaton starting from domain wall initial state for a system with $L = 298$ sites and $N_P = 100$. (Black and white dots correspond to occupied and empty sites). (a) At short times particles spread ballistically into the empty region. Scattering events appear at regular time intervals at the boundaries of the red dashed triangle which defines the region of ballistic behavior. (b) At later times when particle density is lower the constraint becomes more effective, leading to the logarithmic spreading of particles into the empty region. The inset shows the dependence of the current across the domain wall on time that has a clear ballistic regime of linear increase with time followed by slow logarithmic growth at later times.

As each local gate is particle conserving, in order to allow for non-trivial transport, we shift gate position by one site after each layer, as shown in Fig. 10(a). Consequently, the circuit has a 7-layer unit cell in the time direction. Additionally, the order of gate applications is also important, as the gates $U_{1,2}$ generally do not commute with each other. Alternating the layers of $U_1$ and $U_2$ gates proves to be the best choice, as it implements all allowed particle hopping processes, leading to the circuit shown in Fig. 9.

Using this cellular automaton we are able to simulate the time-evolution of very large systems to extremely long times. As the setup implements the same constraint as the Hamiltonian dynamics, we conjecture that it should present similar features. For instance, initializing the system in a dense-empty configuration similar to the $|\text{DW}\rangle$ state, we expect the dense region to spread quickly into the empty one, until eventually it stretches too much and its propagation slows down due to the constraint.

We study the evolution to the domain-wall initial state for a system of $L = 298$ sites and $N_p = 100$ particles. Since this model is deterministic, the density as a function of circuit depth is a binary function, $n_i(t) \in \{0, 1\}$. Figure 10(a) shows the short-time density dynamics ($t < 1000$). We observe ballistic particle transport in the dense regime. On the one hand, the position of the rightmost particle moves to the right. On the other hand, defects (holes) propagate within the dense domain wall state. The simulation reveals notable difference in velocities of holes and spreading of the rightmost particle, that is expected in view of the inversion breaking symmetry within the model.

The ballistic expansion of the particles is followed by a logarithmic slowdown at later times as shown in Fig. 10(b). Much akin to the Hamiltonian dynamics, this slowdown is due to the lower density reached at later times as the front moves to the right and more particles become temporarily frozen due to the constraint. To further probe the two distinct behaviors observed in the cellular automaton, in the inset of Fig. 10(b) we show the time-evolution of the particle flow across the domain wall $\delta n(t)$ as in Eq. (18). From the initial linear behavior, $\delta n(t)$ abruptly enters a logarithmic regime as it exceeds the extent of the ballistic region, corresponding to $i \approx 180$.

The study of the circuit evolution for the domain-wall initial state then shows the overall similar characteristic inhomogeneous dynamics as the quantum system. At early times, and close to the initial domain wall $i = N_p$, the transport of particles and holes is ballistic as for $t \leq 1$ in the quantum case (see Fig. 7). However, as the density spreads and particle density lowers, ballistic spreading is replaced by a logarithmic slow dynamics. We notice, however, that the automaton lacks the super-diffusive plateau observed in the Hamiltonian dynamics.

## 5 Discussion

In this work, we introduced a family of models characterized by a conserved $U(1)$ charge and strong inversion symmetry breaking. We observe that quantum Hilbert space fragmentation [37] in such models can be understood from the recursive construction of special weakly entangled eigenstates coexisting with volume-law entangled eigenstates in the spectrum. In addition, we investigate the dynamics of the system in a quantum quench launched from the domain wall initial state. Although the long time saturation value of particle density is consistent with thermalization, we observe two distinct regimes in particles spreading from the domain wall initial state. An initial superdiffusive particle spreading at high density is dramatically slowed down at lower densities, leading to a logarithmically slow approach of density to its saturation value. While the superdiffudive plateau has an extent in time that increases with system size, its sensitivity to the choice of initial state suggests that it might be related to the particular nature of the domain wall state considered here and may not have a universal hydrodynamic description. The second, slow transport regime, on the other hand, is not so sensitive to the particular initial state and we attribute this to the structure of the constrained Hamiltonian. In addition, we also reproduce the logarithmic dynamics in a classical cellular automaton that features the same symmetries, although at early times the cellular automaton features ballistic dynamics in contrast to slower but still superdiffusive spreading of particles in the Hamiltonian model.

Our work suggests that the interplay of constraints and broken inversion or other spatial symmetries may lead to new universality classes of weak thermalization breakdown and quantum dynamics. In particular, the quantum Hilbert space fragmentation in the considered model gives rise to a number of weakly entangled eigenstates that can be interpreted as quantum many-body scars [47, 48]. The number of these eigenstates scales exponentially with system size. Moreover these eigenstates may be constructed in a recursive fashion, by reusing eigenstates of a smaller number of particles. This is in contrast to the PXP model, where the number of scarred eigenstates is believed to scale polynomially with system size [42, 44], though existence of a larger number of special eigenstates was also conjectured [36].

Although we presented an analytic construction for certain weakly entangled eigenstates and demonstrated their robustness to certain deformations of the Hamiltonian, a formal definition of recursive quantum Hilbert space fragmentation, beyond our phenomenological observation, remains an interesting direction for future work. The complete enumeration and understanding of weakly entangled eigenstates may give further insights into their structure and requirements for their existence. In addition, a systematic study of the emergence of quantum Hilbert space fragmentation in the largest sector of a classically connected Hilbert space in other constrained systems, like the XNOR or the Fredkin models is desirable [7, 12].

From the perspective of particle transport, the numerical data for the dynamical exponent controlling particle spreading suggests that our family of constrained models features superdiffusive dynamics [36, 51–54] from a particular domain wall initial state in the largest connected sector of the Hilbert space. Thus, although it is less robust compared to other examples, understanding and quantifying the emergence of superdiffusion in the present and similar mod-

els with longer range of assisted hopping remains and interesting question. In particular, the models considered in our work may be implemented using quantum simulator platforms using control-swap gates of various ranges. Thus, an experimental study of such models may reveal novel valuable insights into their physics and the universality of their transport phenomena, which are beyond the reach of current state of the art numerical and theoretical approaches.

## Acknowledgments

We would like to thank Raimel A. Medina, Hansveer Singh, and Dmitry Abanin for useful discussions.

**Funding information**    The authors acknowledge support by the European Research Council (ERC) under the European Union's Horizon 2020 research and innovation program (Grant Agreement No. 850899). We acknowledge support by the Erwin Schrödinger International Institute for Mathematics and Physics (ESI).

## A  Thermalization within the largest subsector of the Hilbert space

In order to show the ergodic behavior of the eigenstates of the Hamiltonian, we study the distribution $P(s)$ of the energy differences in the sorted eigenspectrum weighted by the mean level spacing $\Delta$, $s_i = (\epsilon_i - \epsilon_{i-1})/\Delta$. It is known that thermal systems which satisfy the eigenstate thermalization hypothesis are characterized by level statistics in agreement with the prediction of the Gaussian orthogonal ensemble (GOE), $P_{\text{GOE}}(s) = \frac{\pi}{2} s e^{-\frac{\pi}{4} s^2}$.

However, before discussing the level statistics, the discussion of the density of states is in order. The Hamiltonian $\hat{H}_2$ has a spectral reflection property with respect to $E = 0$ and it presents an exponentially large in system size number of zero modes, as highlighted by the peak in the density of states $\rho(0)$ shown in Figure 11(a). The large number of zero energy eigenstates is explained by the bipartite nature of the adjacency graph that describes the Hamiltonian, see Figure 8 for an example. In a bipartite graph there exist two sets of nodes $\mathcal{P}_{1,2}$ labeled by different product states, such that the action of the Hamiltonian on states belonging

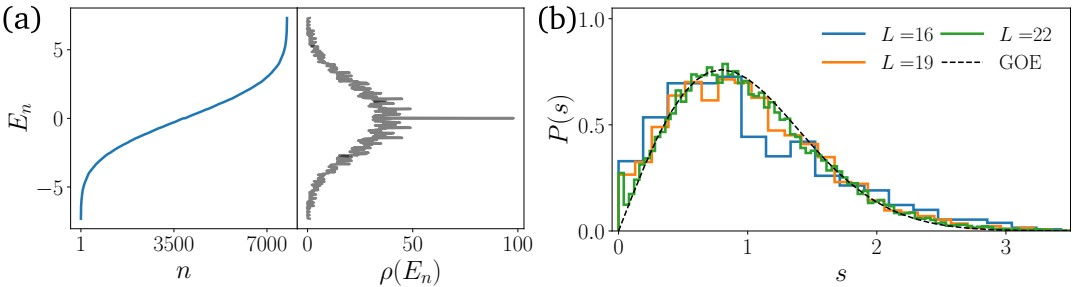

Figure 11: (a): As shown in the left sub-panel, the spectrum is symmetric with respect to $E_n = 0$, such that for any eigenstate with eigenvalue $E_n$ there is a second state with energy $-E_n$. Additionally, the model has a large number of zero energy eigenstates, as highlighted by the peak of the density of states $\rho(E_n)$ in the right sub-panel. We show data for $N_p = 7$ and $L = 19$. (b): The level spacing distribution $P(s)$ shows good agreement with the GOE prediction, shown as a black dashed line, thus confirming the presence of level repulsion within the largest subsector.

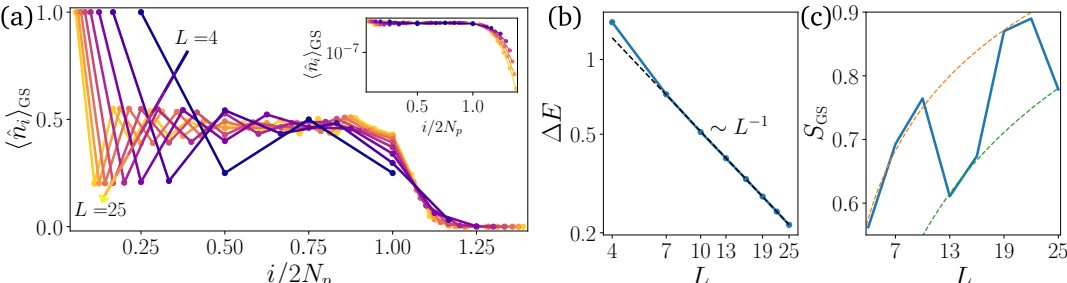

Figure 12: (a): The density profile of the ground state $\langle \hat{n}_i \rangle_{\text{GS}}$ shows large particle occupation up to $i = 2N_p$. Outside this region, the density starts decaying exponentially, as shown in the inset. (b): The finite size scaling of the energy gap $\Delta E$ shows that it vanishes as $1/L$, thus indicating that the model is gapless in the thermodynamic limit. (c): Entanglement entropy across the central cut grows logarithmically with strong finite size corrections (dashed orange and green lines show logarithmic fits), providing additional evidence that the ground state is critical.

to the set $\mathcal{P}_1$ yields a state in the set $\mathcal{P}_2$ and vice versa. These two partitions are identified by the eigenvalue of the parity operator $\hat{\mathcal{P}} = \prod_j (1 - 2\hat{n}_j)^j = \prod_j (-\sigma_j^z)^j$, where $\sigma_j^z = 2\hat{n}_j - 1$ is the corresponding Pauli matrix. It is known that a bipartite graph has a number of zero modes bounded from below by the difference in the size of the two sets $P_1$ and $P_2$ [55].

In fact, when the two partitions have a different number of states, a non-trivial solution of the Schrödinger equation for a zero energy eigenstate can be expressed as a system of $n_1$ linear equations for $n_2$ variables. If $n_2 > n_1$, there are at least $n_2 - n_1$ linearly independent solutions. In this case, in spite of the bound not being tight, both the number of zero modes and the lower bound from the bipartite structure of the graph describing the Hamiltonian increase exponentially with system size, albeit with different prefactors in the exponent. This suggests that the present understanding of the zero mode subspace is incomplete, inviting further research. In particular, using the disentangling algorithm [56] may give valuable insights. This may also help to develop a more complete understanding of the recursive Hilbert space fragmentation, since its mechanism relies on the zero energy eigenstates with vanishing particle density on the last sites of the system, see Section 3.2.

In Figure 11(b) we show the level spacing distribution for $L \in [16, 22]$ in the interval $[E_{\text{GS}}, -0.1]$, where $E_{\text{GS}}$ corresponds to the ground state energy. Note that due to the spectral reflection property of the Hamiltonian, taking into account only negative energies yields the same results as considering the whole spectrum. To obtain $P(s)$, we unfold the spectrum in the given interval through polynomial interpolation of the integrated density of states. The agreement with the GOE prediction suggests that despite the presence of a constraint, the levels develop repulsion within the largest connected sector of the Hilbert space and the model is not integrable.

## B  Ground state characterization

In this Appendix we characterize the ground state, studying the scaling of the energy gap and of the entanglement entropy. As the Hamiltonian (3) only has hopping terms, the low lying eigenstates need to have a large overlap with product states that maximize a number of configurations to which hopping is allowed to. In graph language, see Figure 8 for an example, these product states correspond to vertices with the largest possible connectivity. For $r = 2$, the

state with highest connectivity is $|\underbrace{\bullet\circ\bullet\circ\bullet\cdots\bullet}_{2N_p}\underbrace{\circ\circ\circ\circ\cdots\circ}_{L-2N_p}\rangle$, with connectivity $2N_p-1$, hence we expect the ground state to have a large weight on the initial $2N_p$ sites. In Figure 12(a) we plot the density profile of the ground state of the Hamiltonian (2) for different system sizes from $L=4$ to $L=25$ against a rescaled $x$-axis $i/2N_p$. The figure confirms the prediction, the ground state is confined within the first $2N_p$ sites, with an exponentially decaying density outside of this region, as shown in the inset. This behavior is different from the one observed in the quantum East model in absence of particle conservation [8,14], where occupation immediately decays exponentially.

We further study the scaling of the energy gap and of the entanglement entropy. As clearly shown in Figure 12(b), the energy gap $\Delta E$ vanishes as the inverse system size, suggesting that model is in a gapless phase in the thermodynamic limit. Additionally, the entanglement entropy of the ground state across the central cut in the chain presents a slow logarithmic growth. These results suggest that the ground state is critical.

# C  Construction of left parts of separable eigenstates

In this section we report the left-restricted eigenvectors entering Eq. (11) for all sub-system sizes we were able to investigate numerically for $r=2$. These were used in the main text to correctly count the global number of zero entanglement eigenstates $\mathcal{N}_S$ shown in Figure 3(b). We remind here that these eigenstates have to fulfill two conditions

(i)  they have to be an eigenstate on the problem restricted to $m$ particles in $\ell$ sites, with $\ell \leq 3m-2$.

(ii)  They must have zero density on the boundary site $\ell$: $\langle \psi_m^\ell | \hat{n}_\ell | \psi_m^\ell \rangle = 0$.

Additionally we observe that these left-restricted eigenvectors always correspond to zero energy.

To obtain these states, we take advantage of the large number of zero modes of the Hamiltonian (2). Within the degenerate sub-space, one can perform unitary transformations and obtain a new set of zero energy eigenstates where at least one satisfies the condition (ii) above. To find the correct states in an efficient way, we build the matrix $N_{\alpha,\beta} = \langle E_\alpha^{m,\ell} | \hat{n}_\ell | E_\beta^{m,\ell} \rangle$ of the expectation values of the density on the last site on eigenstates of the Hamiltonian reduced to $(m,\ell)$. We then diagonalize $N_{\alpha,\beta}$ and check whether it has zero eigenvalues. If so, the corresponding eigenvector is still an eigenstate of the reduced Hamiltonian, and, by construction, it satisfies condition (ii). We notice that this method implements a sufficient condition, which implies that there could be other states that fulfill the same set of restrictions. However, our goal here is merely to provide evidence of existence of these states in several different system sizes. In the following, we list the states for $m=3,4,5$ and $\ell=6,9,11$ respectively.

$$
\begin{aligned}
|\psi_3^6\rangle =& \frac{1}{\sqrt{2}}\big(|\bullet\bullet\circ\circ\bullet\circ\rangle - |\bullet\circ\bullet\bullet\circ\circ\rangle\big), \\
|\psi_4^9\rangle =& \frac{1}{2}\big(|\bullet\bullet\circ\circ\bullet\circ\circ\bullet\circ\rangle - |\bullet\bullet\bullet\circ\circ\circ\circ\bullet\circ\rangle\big) + \frac{1}{4}\big(|\bullet\circ\circ\bullet\bullet\bullet\circ\circ\circ\rangle + |\bullet\circ\bullet\bullet\circ\bullet\circ\circ\circ\rangle \\
& + |\bullet\circ\circ\bullet\circ\bullet\bullet\circ\circ\rangle + |\bullet\bullet\bullet\circ\circ\bullet\bullet\circ\circ\circ\rangle - |\bullet\bullet\circ\bullet\bullet\circ\circ\circ\circ\rangle - |\bullet\bullet\circ\circ\bullet\bullet\circ\circ\rangle \\
& - |\bullet\circ\circ\bullet\bullet\circ\circ\circ\bullet\circ\rangle - |\bullet\circ\bullet\circ\bullet\circ\circ\bullet\circ\rangle\big), 
\end{aligned}
$$
$$
\begin{aligned}
|\psi_5^{11}\rangle =& \frac{1}{\sqrt{6}}\big(|\bullet\circ\circ\bullet\bullet\bullet\bullet\circ\circ\circ\circ\rangle + |\bullet\circ\bullet\bullet\circ\circ\bullet\bullet\bullet\circ\circ\circ\rangle + |\bullet\bullet\circ\circ\bullet\bullet\bullet\circ\circ\bullet\circ\circ\rangle \\
& + |\bullet\bullet\bullet\circ\circ\circ\circ\bullet\circ\circ\bullet\circ\rangle - |\bullet\circ\circ\bullet\circ\bullet\bullet\circ\circ\circ\circ\rangle - |\bullet\bullet\bullet\circ\circ\bullet\circ\circ\bullet\circ\circ\rangle\big).
\end{aligned}
$$

(C.1)

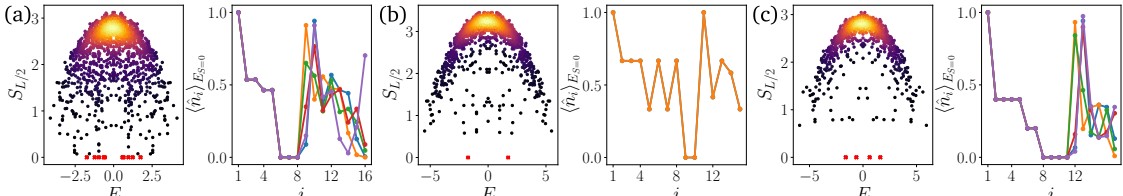

Figure 13: (a): entanglement entropy of the eigenstates of the Hamiltonian for range $r = 2$, random hopping parameters $t_1 = 0.84$, $t_2 = 0.49$ , and system size $L = 16$. The presence of zero entanglement eigenstates, highlighted by the red crosses, confirms that quantum fragmentation is insensitive to the value of the hopping amplitudes. (b)-(c): A similar result is obtained for different values of the range $r$. The central panels refer to $r = 1$, $N_p = 8$ and $L = 15$, while the right ones show $r = 3$, $N_p = 5$ and $L = 17$.

Additional states are present, which we do not write down for the sake of brevity. However, we point out the existence of recursively stacked eigenstates, as mentioned in the main text, and of states where the right part corresponds to a single isolated particle.

## D  Quantum Hilbert space fragmentation for generic Hamiltonian parameters

Throughout the main text, we often mentioned that the results regarding quantum fragmentation hold irrespective of the range of the constraint $r$ and of the values of the hopping amplitudes $t_\ell$. In the following, we provide evidence in support of the generality of recursive fragmentation.

In Figure 13, we first show the entanglement entropy of eigenstates for $r = 2$ and Hamiltonian

$$\hat{H} = \sum_{i=2}^{L-1} (t_1\hat{n}_{i-1} + t_2\hat{n}_{i-2} - t_2\hat{n}_{i-1}\hat{n}_{i-2})\left(\hat{c}_{i+1}^\dagger \hat{c}_i + \text{H.c.}\right), \tag{D.1}$$

with generic, although homogeneous, hopping amplitudes $t_1, t_2$. In the leftmost panel, we highlight the presence of zero entanglement eigenstates in the half-chain cut for a random choice of the hopping parameters. The density profile of these special eigenstates is similar to the one showed in Figure 3(a), although the density profile in the left region has more complicated pattern due to the different values of $t_{1,2}$.

Next, we show the presence of recursive fragmentation in the generic Hamiltonian (3). In the central and right panels of Figure 13 zero entanglement eigenstates (red crosses) appear across the central cut for both $r = 1$ and $r = 3$. As for the random $t_{1,2}$ case, the structure of these eigenstates is akin to the one obtained in Eq. (11), featuring an empty region of $r + 1$ sites disconnecting the left region from the right one. Thus we provide numerical evidence in support of the generic form of the zero entropy eigenstates $|E_{S=0}\rangle$ proposed in the main text.

## E  Transport in different initial states

In the main text, we discuss the transport properties of the domain wall initial state, observing an unexpected superdiffusive behavior. Although at a finite time $t^*$, the dynamics slow down, showing signatures of logarithmically slow transport, the linear increase of $t^*$ with system size

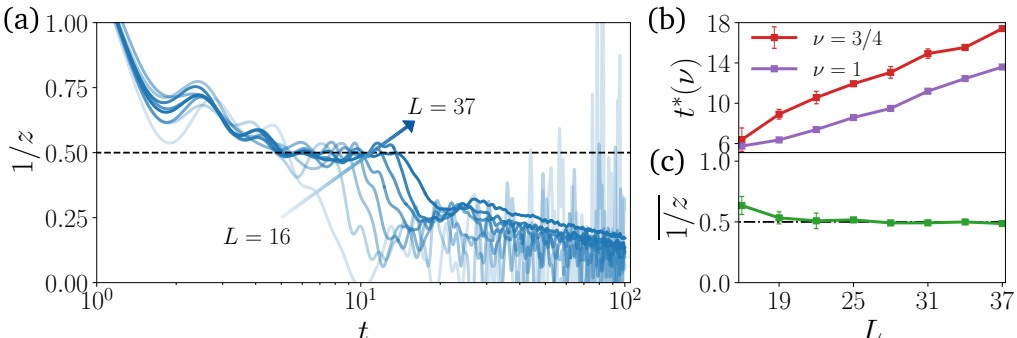

Figure 14:     (a) The inverse dynamical exponent for different system sizes ($L \in [16, 37]$ from more to less opaque) averaged over 10 random initializations. The behavior of $1/z$ is qualitatively similar to the one observed for the $|\text{DW}\rangle$ initial state, although the plateau is compatible with diffusion in this case. (b) The onset time of the logarithmic behavior, $t^*(\nu)$, averaged over the different choices of $|\psi_0\rangle$ ($\nu = 3/4$) and for the domain wall initial state ($\nu = 1$), shows a clear increase with system size. (c) The average value of the inverse dynamical exponent within the plateau $\overline{1/z}$ shows an initial decrease with system size, before eventually saturating to a value compatible with $1/z = 0.5$ (black dashed line).

suggests that this feature persists in the thermodynamic limit. In this Appendix, we explore the dynamics of random initial states with varying density to understand the generality of the dynamics observed in the $|\text{DW}\rangle$ initial state.

To this end, we initialize the system in a random superposition of all product states $|\varphi_i\rangle$ with average particle density $\nu = 3/4$ in the leftmost sites

$$|\psi_0\rangle = \frac{1}{N} \sum_i c_i e^{\iota \phi_i} |\varphi_i\rangle \,, \tag{E.1}$$

where the amplitude $c_i$ and the phase $\phi_i$ are drawn randomly from a uniform distribution in $[0, \pi]$ and $N$ is the normalization factor. After running the dynamics up to time $t = 100$ for 10 different random initial states, we obtain the inverse dynamical exponent $1/z$ as in the main text and average among the different states.

As shown in Figure 14, the inverse dynamical exponent presents a behavior qualitatively similar to the one observed for the $|\text{DW}\rangle$ initial state, with a plateau extending up to time $t^*(\nu)$ before eventually slowing down to a logarithmic behavior. In this case, however, the plateau suggests diffusive dynamics, suggesting the possibility of a density-dependent transport exponent. In panel (b) of the same Figure, we show the average $t^*(\nu)$ as a function of system size, clearly showing that the plateau becomes longer as $L$ increases both for the $|\text{DW}\rangle$ initial state ($\nu = 1$) and for the present case. Finally, we study the value of the dynamical exponent within the plateau, $\overline{1/z}$, by averaging $1/z(t)$ in a time window $[t_0(L), t_1(L)]$ for each realization of the initial state. The values of $t_0(L)$ and $t_1(L)$ are given in the following table.

| $L$ | $t_0$ | $t_1$ | $L$ | $t_0$ | $t_1$ |
|-----|-------|-------|-----|-------|-------|
| 16 | 1 | 5 | 28 | 5 | 10 |
| 19 | 2.5 | 7.5 | 31 | 5 | 12 |
| 22 | 3 | 8 | 34 | 5 | 13 |
| 25 | 4 | 9.5 | 37 | 5 | 13 |

As shown in panel (c), after a decrease with system size at small values of $L$, the average inverse dynamical exponent $\overline{1/z}$ among different $|\psi_0\rangle$ realizations stabilizes to a value compatible with

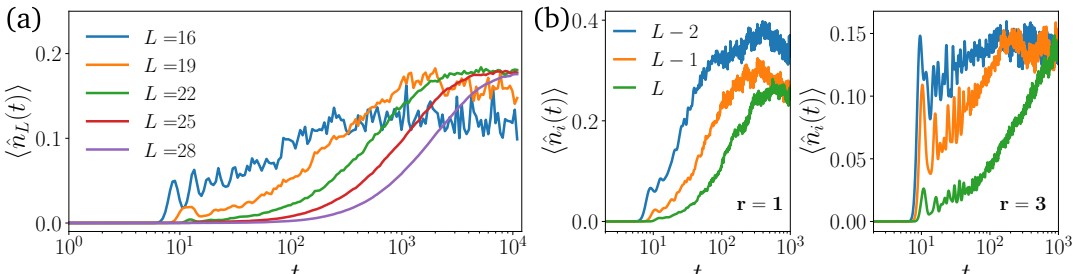

Figure 15: (a): Dynamics of the density on the last site $\langle \hat{n}_L(t) \rangle$ for several different system sizes. The slow logarithmic growth is evident for all $L \geq 16$. At larger system sizes $L \geq 19$ the slope becomes independent of system size, as well as the saturation value, thus suggesting a universal behavior. (b): Density dynamics for different values of the range $r$ always show logarithmic behavior. While the quantitative details change between different values of $r$, the qualitative feature of the logarithmic growth is a constant. The data are obtained on a chain of $N_p = 9$ and $L = 17$ for $r = 1$, and $N_p = 6$ and $L = 21$ for $r = 3$.

diffusion $1/z = 0.5$. We also notice that the small standard deviation, represented by the error bars, suggests that this behavior is typical among the studied states.

## F Dynamics of the domain wall initial state for different values of $r$

In the main text we provided evidence of slow dynamics from the time-evolution of the density operator in large systems and from the behavior of the root-mean-square displacement. Here, we present some additional data regarding system size scaling of the density dynamics as well as the observation of slow dynamics for generic $r$. Finally, we present an additional measure for the logarithmic behavior of the particles spreading.

In Figure 15(a) we show the system size scaling of the dynamics of the density on the last site of the chain, $\langle \hat{n}_L(t) \rangle$. All the curves present logarithmic growth, and for larger system sizes $L \geq 19$ the slope becomes roughly constant. The absence of logarithmic behavior for smaller system sizes $L < 16$ is in agreement with the data shown in the main text, where $R(t)$ quickly saturates for $L = 13$.

Similar slow dynamics are observed in the time-evolution generated by Hamiltonians with generic constraint range $r$. In Figure 15(b) we present the growth of the density in the last three sites of two chains of length $L = 17$ and $L = 21$ for $r = 1$ and $r = 3$ respectively. As the data suggest, the dynamics in the rightmost part of the chain always presents logarithmic behavior, irrespective of the range of the constraint. However, the quantitative details are affected by $r$.

To analyze the spreading of the density, in the main text we presented the behavior of the root-mean-square displacement $R(t)$ together with the respective dynamical exponent $z_R(t)$. Here, we approach the same question using a different measure, namely the time-dependence of the expansion of the density profile. This spreading distance $\delta r$ is defined as the distance from the domain wall boundary, $i = N_p$, at which density becomes larger than a certain threshold $\varepsilon \ll 1$. The spreading distance $\delta r$ is expected to asymptotically behave as a power-law in time, defining a dynamical exponent $z_r$ such that $\delta r \approx t^{1/z_r}$. However, the limited system sizes available to our numerical study do not allow us to reach the asymptotic regime, and we are forced to study the time-dependent analogue $z_r(t)$, obtained through the logarithmic

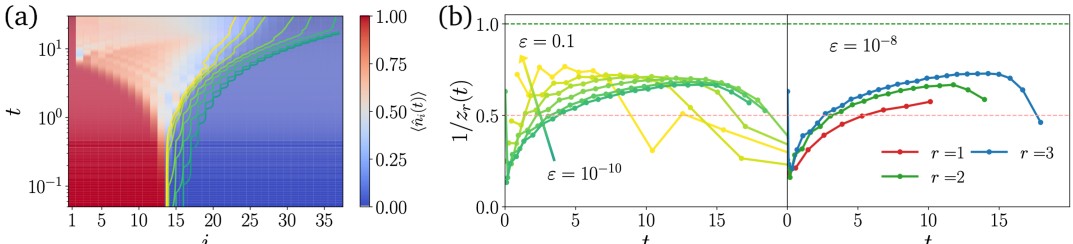

Figure 16: (a) Spreading of the density in a system with $L = 37$ sites and $N_p = 13$ bosons. Lines of constant value $\varepsilon$ highlight the very different behavior observed in the two regions $i \lessgtr 2N_p$. (b) The inverse dynamical exponent $1/z_r(t)$ is always super-diffusive. While for a large threshold it decays to 0 indicating the onset of logarithmic growth, for small values of $\varepsilon$ the dynamical exponent seems to saturate approaching the asymptotic value (weakly dependent on the threshold value), before the onset of boundary effects. As shown in the right panel, the asymptotic $1/z_r$ is super-diffusive behavior is generic irrespective of the choice of the range of the constraint. The data shown in this panel correspond to $N_p = 11$ and $L = 21, 31, 41$ for $r = 1, 2, 3$ respectively.

derivative of the spreading distance with respect to time, $(z_r(t))^{-1} = d \ln \delta r / d \ln t$.

In panel (a) of Figure 16 we show a heat-map of the density dynamics for $L = 37$ sites, superimposed with curves of constant $\langle \hat{n}_i(t) \rangle = \varepsilon$, for values of $\varepsilon \in [0.1, 10^{-10}]$, above the accuracy limit $O(10^{-12})$ of the 4-th order Runge-Kutta algorithm. For each threshold, we show in panel (b) the time-dependent dynamical exponent. For the largest values of $\varepsilon$ the dynamical exponent has a super-diffusive plateau at $1/z_r(t) \approx 0.7$ before quickly vanishing as expected from the logarithmic dynamics of the density. On the other hand, at smaller thersholds the dynamical exponent seems to saturate to a finite value, before it eventually starts decreasing due to boundary effects.

The saturation value of the time-dependent dynamic exponent for small thresholds has a weak dependence on the value of $\varepsilon$. As $\varepsilon \to 0$, $1/z_r$ approaches a $r$-dependent saturation value, monotonically increasing as the range of the constraint becomes larger, as shown in the right panel of Figure 16(b). This behavior is in agreement with the expectation that at $r \to \infty$ the system should approach ballistic dynamics.

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
