# Peer review of "Hilbert space fragmentation and slow dynamics in particle-conserving quantum East models"

_SciPost Physics, doi:SciPost Phys. 15, 093 (2023)_

## Round 1 · Referee Report · Anonymous (Referee 1) · 2022-12-18

Report

The authors investigated Hilbert-space fragmentation and slow dynamics in a U(1)-symmetric quantum East model. The kinetic constraints give rise to exponentially many disconnected subsectors on top of those labeled by the U(1) quantum number. Furthermore, by studying transport within a given subsector instead of averaging over the full Hilbert space, the authors discovered superdiffusive behavior, in contrast to subdiffusion that was reported previously.

I find that the results reported in this work are quite interesting, and constitute a valuable addition to the recent line of research on quantum dynamics in kinetically constrained models. Therefore, I recommend publication in SciPost Physics.

I only have a few questions: 1. In the construction of eigenstates with zero entanglement Eq. (11), the notation suggests that the states $|\psi_m^l\rangle$ and $|\psi_R\rangle$ are not quite on equal footing. It seems that if I replace $|\psi_R\rangle$ with $|\psi_{m'}^{l'}\rangle$, it still leads to a valid eigenstate with zero entanglement. Is that right? Can I think of $|\psi_m^l\rangle$ as a special case of $|\psi_R\rangle$ with the additional condition that the occupation number on the rightmost site must be zero?

  1. I'm wondering whether the authors have looked at transport properties restricted to other subsectors. Does any of these subsectors show diffusive and/or subdiffusive behavior?
  • validity: -
  • significance: -
  • originality: -
  • clarity: -
  • formatting: -
  • grammar: -

Author:  Pietro Brighi  on 2023-01-31  [id 3290]

(in reply to Report 1 on 2022-12-18)

We thank the referee for their careful read of our manuscript and positive comments. Regarding their questions, in the following we provide replies to clarify their doubts.

1) It is true that replacing $\psi_R$ with an eigenvector of the form$ \psi_{m’}^{l’}$ leads to a zero entanglement eigenstate, with eventually an additional zero entanglement cut. This is precisely the mechanism causing the recursive fragmentation we observe. One can indeed think of $\psi_m^l$ as a special case of $\psi_R$ with zero density on the rightmost site. What is important however is that the opposite statement does not hold: one can not replace $\psi_m^l$ with $\psi_R$, which is why the two are not treated on equal footing.

2) We thank the referee for this insightful comment. Note that both classical and quantum fragmentation will lead to a left part of the chain becoming disconnected from the remainder. In case of classical fragmentation that would lead to similar transport properties but for a smaller system size. For quantum fragmentation the left segment will be an eigenstate and thus trivial. Therefore in both situations we expect to have qualitatively similar transport properties to those observed in the largest connected sector.

---

## Round 1 · Referee Report · Anonymous (Referee 2) · 2023-2-2

Report

In their manuscript, the authors introduce a particle number conserving version of the 'quantum east model', which is itself a quantum generalization of the so-called east model, one of the simplest examples of a facilitated / kinetically constrained spin model. In their version, they consider a one-dimensional chain of hard-core bosonic particles, which are allowed to hop between neighboring sites i,i+1 only if at least one site in the interval [i-r,i-1] is already occupied by a particle, mostly focusing on the case with r=2.

They find a variety of interesting phenomena in these models. Some of these involve the structure of Hilbert space, which fragments into many dynamically disconnected sectors. First, they argue that in a symmetry sector with N particles, there are exponentially many in N subsectors that can be labeled by conserved quantities that are diagonal in the particle-number basis; these provide a case of what has been termed classical fragmentation. Furthermore, they argue that there is additional fragmentation beyond these, finding an exponential (in system size) eigenstates that have zero entanglement over specific bipartitions of the chain but are not overall product states. The authors interpret these as pointing towards quantum Hilbert space fragmentation, as previously defined in Ref. 37. They provide a recursive construction for such eigenstates and term this structure 'recursive Hilbert space fragmentation'.

The authors also study dynamical features of their models. They consider the melting of a domain wall configuration and characterize it through a quantity they refer to as the mean-square displacement (MSD); they find unusual behavior, with a super-diffusive (but sub-ballistic) melting, which at a later time gives way to a very slow, logarithmic spreading. In order to better understand this behavior, they study a classical, deterministic cellular automaton version of their model, and they find that it captures some of the behavior, namely the late-time logarithmic regime, while differing from the quantum model at early times, where it shows a ballistic behavior.

The paper thus presents a number of interesting observations which certainly show that the number-conserving east models provide a useful venue for exploring questions of kinetically constrained dynamics. As such, it is well suited for publication in SciPost. However, I found many of the explanation in the paper quite unclear and it left me with the impression that many of these features, and their relationship to each other, are not really understood. I think some of these questions should be clarified before publication. I will detail these questions below.

The first issue concerns the nature of Hilbert space fragmentation. In the manuscript, there is some discrepancy between the way that the classical and quantum fragmentations are discussed: while the former is treated by constructing explicitly conserved quantities, the discussion of the latter focuses on the form of eigenstate wavefunctions. This makes the interpretation somewhat difficult. In particular, connecting these latter observations to the definitions in Ref. [37] would be very helpful. There, fragmentation defined in terms of conserved quantities, more precisely the commutant algebra made up by all operators that commute with each individual term in H. What is the claim implied on the commutant algebra by the observation of disentangled eigenstates? Alternatively, is there a basis in which a block-diagonal structure (within the already resolved 'classically framented' blocks) can be demonstrated?

The precise relationship between the observed transport behavior and the fragmented structure also seems unclear. For example some features are reproduced by a classical model while others are not - does that mean that the former originate from classical fragmentation while the latter have to do with quantum fragmentation in some way? The ballistic spreading in the classical cellular automaton model raises the possibility that it might be integrable; has that been ruled out? To probe physics that originates only from the local kinetic constraints, a more natural comparison would be provided by stochastics cellular automata (as e.g. in Ref. 10) which tend to show generic hydrodynamic features. The authors do offer an explanation of the late-time logarithmically slow spreading in terms of Hilbert space structure. I found this rather qualitative and a bit hard to follow: it was unclear whether this was only meant as a rough intuition or something more rigorous. In any case, the behavior it purports to explain is a finite size effect, occuring at time t ~ L. When considering transport properties, one takes the thermodynamic limit first; so the more interesting observation os the super-diffusive regime which, as far as I can tell, remains unexplained.

There are some other aspects of the discussion of transport that I found unclear. First, the quantity the authors study - a mean-square displacement for a domain wall melting - seems unusual. Has the same quantity been used to probe transport in other systems? More familiar probes would be either the total charge transported through the domain wall or a MSD defined in terms of dynamical charge-charge correlations, both of which can be related to the current operator. The latter, in particular, raises the question whether the observed anomalous relaxation is specific to the domain wall initial state, or if it remains even when one considers correlations averaged over different initial states. Furthermore, I wanted to clarify whether the authors expect the dynamical exponent to be tunable by the range r of interactions. This claim I would find very surprising as such exponent tend to be well explained by coarse-grained hydrodynamical theories that should be insensitive to such details.

A few smaller questions and comments I had:

  • How were the simulations of the quantum model performed? The manuscript contains data for systems of length L = 37 which is larger than can usually be obtained in exact numerics. Are the numerics making use of the fragmented Hilbert space?

  • Is there a precise definition of 'recursive fragmentation'? While there is a concrete construction in the paper, it was not clear to me what the authors mean by the term more generally.

  • I found some of the discussions regarding fragmentation hard to follow. For example, while Eq. (8) is clear enough, I would not have been able to write it based on the description that precedes it.

  • validity: -
  • significance: -
  • originality: -
  • clarity: -
  • formatting: -
  • grammar: -

Author:  Pietro Brighi  on 2023-03-08  [id 3452]

(in reply to Report 2 on 2023-02-02)

We thank the Referee for the thorough report which has helped us improve the presentation of our work. We provide our answers to the Referee’s questions below.

"The first issue concerns the nature of Hilbert space fragmentation. In the manuscript, there is some discrepancy between the way that the classical and quantum fragmentations are discussed: while the former is treated by constructing explicitly conserved quantities, the discussion of the latter focuses on the form of eigenstate wavefunctions. This makes the interpretation somewhat difficult. In particular, connecting these latter observations to the definitions in Ref. [37] would be very helpful. There, fragmentation defined in terms of conserved quantities, more precisely the commutant algebra made up by all operators that commute with each individual term in H. What is the claim implied on the commutant algebra by the observation of disentangled eigenstates? "

We agree with the Referee that the presentation was inconsistent. To resolve this inconsistency, we will add the operators relating to the quantum fragmentation to the manuscript, which take a similar form as those pertaining to the classic fragmentation

$$ \hat{O}{\psi_m^\ell}^{q} = \hat{\mathcal{P}}}\Bigr[\prod_{k=\ell+1}^{\ell+q+1} (1-\hat{nk)\Bigr] \hat{n}, $$
where $\hat{\mathcal{P}}_{\psi_m^\ell}$ is a projector on the special eigenstate $|\psi_m^\ell\rangle$. Furthermore, we will change the manuscript to discuss the relation of the algebra generated by these operators to the commutant algebra introduced in Ref. [37]. Note however, that the commutant algebras refer to operators that commute with each individual term of the Hamiltonian, while in our case the operators commute only with the full Hamiltonian and not with the individual terms. Nevertheless, both the quantum and classical operators form closed algebras whose size scales exponentially with system size. This exponential scaling arises from the possibility of composing operators acting on different parts of the chain and yielding new fragmented sectors. We will add this explanation to Section 3.2 (page 8). We believe that the modifications to the manuscript make the presentation clearer and hope that it answers the Referee’s questions.

"Alternatively, is there a basis in which a block-diagonal structure (within the already resolved 'classically fragmented' blocks) can be demonstrated?"

Following the question by the Referee, we will define the basis needed to observe the block diagonal structure explicitly in Sec. 3.2. Since the operators relating to quantum fragmentation are simply projectors onto eigenstates of a subsystem padded by a sufficient number of empty sites, a natural basis in which the Hamiltonian becomes block-diagonal is one which includes these particular padded eigenstates. For example taking a basis formed by $|\psi_\ell^m\rangle\otimes|00\rangle\otimes|\phi\rangle$ and $|\chi\rangle\otimes|\phi\rangle$ where ${|\phi\rangle}$ forms a basis of the right part of the chain, $(\langle\psi_\ell^m|\otimes\langle00|)|\chi\rangle=0$ and ${|\psi_\ell^m\rangle}\cup {|\chi\rangle}$ form a basis of the left side of the chain.

"The precise relationship between the observed transport behavior and the fragmented structure also seems unclear. For example some features are reproduced by a classical model while others are not - does that mean that the former originate from classical fragmentation while the latter have to do with quantum fragmentation in some way?"

We agree with the Referee that fragmentation can have a significant effect on transport in certain situations. However, in our study we focus on transport within the largest classical and quantum fragments, hence fragmentation has no direct effect on the transport we observe. If that were not the case, one could expect the fragmentation to lead to some sort of suppressed transport as the structure of the fragments would cause the system to thermalize within the effectively disconnected parts of the chain. Nevertheless, we agree that fragmentation can still affect transport indirectly, given that these are both properties stemming from the same Hamiltonian. Indeed, studying the information one could extract about transport from the understanding of fragmentation poses an extremely interesting question. However, we believe that this highly non-trivial problem is beyond the scope of the current work.

"The ballistic spreading in the classical cellular automaton model raises the possibility that it might be integrable; has that been ruled out? To probe physics that originates only from the local kinetic constraints, a more natural comparison would be provided by stochastics cellular automata (as e.g. in Ref. 10) which tend to show generic hydrodynamic features."

We agree that at a quick glance the classical cellular automaton may look integrable, due to clear soliton-like behavior that is seen at short times. However, we note that this is only the case at short times where the constraint has no effect (for the particular domain wall initial state). Hence the particles are effectively free. The picture quickly changes once the density drops sufficiently. Furthermore, the scattering between these excitations does not appear to factorize as shown in the example in Figure 1 of the attachments. This leads us to believe the model is in fact not integrable, but only seems so at short times for the domain wall initial state.

We have also looked at a stochastic cellular automaton. In this case, we initialize the system in the DW initial state and perform time-evolution as follows. First, we select a random site among the occupied ones at time t. We then choose whether to move left or right based on the outcome of a binary random variable that can take values 0 and 1. Finally, if the hopping is allowed by the constraint, we update the state at time t+1 by moving the particle in the chosen direction. If instead the hopping is not allowed, the state at time t+1 remains the same as the one at time t. In this setup, after averaging over 5x10^4 different realizations of the dynamics, we observe clear diffusion at early times, followed eventually by a much slower logarithmic behavior, as the average density decreases (see Figure 2 in the attachments). Due to the large length of the paper, we decided not to include this data into the manuscript.

"The authors do offer an explanation of the late-time logarithmically slow spreading in terms of Hilbert space structure. I found this rather qualitative and a bit hard to follow: it was unclear whether this was only meant as a rough intuition or something more rigorous. In any case, the behavior it purports to explain is a finite size effect, occuring at time t ~ L. When considering transport properties, one takes the thermodynamic limit first; so the more interesting observation os the super-diffusive regime which, as far as I can tell, remains unexplained."

Our explanation of the observed late-time logarithmically slow dynamics was indeed meant to provide an intuition. We will explicitly mention that our explanation is qualitative in the revised manuscript.

Furthermore, we agree that the super-diffusive regime observed in Fig. 7 is extremely interesting and unexpected in a chaotic constrained model and indeed remains unexplained. However, we note that while it is true that the logarithmically slow dynamics appear at t~L in this case, this is not expected to be the case in a generic state (or indeed at high temperature). Instead, we expect the logarithmic dynamics to be present whenever the density of particles drops sufficiently for the constraint to become relevant and block particles from moving to the right for extended periods of time. As shown in Figure 3 of the attachments, random initial states with average density 1/2 in the leftmost part of the chain present a behavior compatible with immediate logarithmic transport (dashed black line). Unfortunately, strong fluctuations preclude us from reaching a definite conclusion about late time transport. A study of larger systems with matrix product state methods may help to resolve this issue. However this would require an efficient way to project the dynamics onto the largest fragment, we are currently not aware of a simple MPO representation of the projectors needed to do so in this model and believe that this is beyond the scope of the present work.

"There are some other aspects of the discussion of transport that I found unclear. First, the quantity the authors study - a mean-square displacement for a domain wall melting - seems unusual. Has the same quantity been used to probe transport in other systems? More familiar probes would be either the total charge transported through the domain wall or a MSD defined in terms of dynamical charge-charge correlations, both of which can be related to the current operator. The latter, in particular, raises the question whether the observed anomalous relaxation is specific to the domain wall initial state, or if it remains even when one considers correlations averaged over different initial states. Furthermore, I wanted to clarify whether the authors expect the dynamical exponent to be tunable by the range r of interactions. This claim I would find very surprising as such exponent tend to be well explained by coarse-grained hydrodynamical theories that should be insensitive to such details."

We agree with the Referee that the quantity that we used in our work is non-standard. Following the suggestion by the Referee, we will provide the data for transferred charge in place of MSD data (Fig.7) which shows qualitatively similar behavior with less noise (see Figure 4 of the attachments).

We agree with the Referee that the observed anomalous behavior could indeed be due to the choice of the initial state. To verify this, we performed additional simulations initialized in different states, which will be presented in an additional Appendix. The transport properties deviate from the superdiffusion observed in the DW initial state, suggesting that, depending on the initial density of particles in the leftmost part of the chain, different behaviors may be observed (see Figure 5 of the attachments for the $\nu=3/4$ case). We believe that this observation is interesting and deserves a more thorough inspection that goes beyond the scope of the present manuscript. We hope that we may be able to address these questions in our future work. Finally, we will revise the manuscript (in the Introduction, in Section 4.2 and in the Discussion) to clarify that the superdiffusive behavior observed is specific to the domain wall initial state.

We agree with the Referee, that the dependence of the transport exponent on the range of the constraint is incompatible with a coarse-grained description, and we will mention this in the discussion. Following the question raised by the Referee, we performed additional simulations for other ranges of the constraint shown in Figure 6 of the attachments. Unfortunately for $r=1$ the timescales we are able to achieve are too short and for $r=3$ the Hilbert space dimension grows too quickly. Consequently, our results about other constraint ranges are less clear.

"A few smaller questions and comments I had: - How were the simulations of the quantum model performed? The manuscript contains data for systems of length L = 37 which is larger than can usually be obtained in exact numerics. Are the numerics making use of the fragmented Hilbert space?"

The simulations were performed using a 4-th order Runge-Kutta integrator to directly integrate the Trotterized time evolution of a state. The simulations were restricted to the largest classical fragment at a fixed number of particles. We did not however make use of the quantum fragmentation. Still, this was enough to significantly reduce the dimensionality of the relevant Hilbert subspace. Namely, for $L=37$ the relevant subspace has a dimension of only $\sim 3\times10^8$.

"- Is there a precise definition of 'recursive fragmentation'? While there is a concrete construction in the paper, it was not clear to me what the authors mean by the term more generally."

The term “recursive fragmentation” in our case refers to the specific construction that we present in our work which explains the observed zero-entanglement eigenstates. Of course, it would be interesting to generalize this notion and define it rigorously, but we feel that this is beyond the scope of the present work. In order to emphasize that our “recursive Hilbert space fragmentation” is more of a phenomenological observation, rather than a rigorously defined notion we will explicitly mention the question of generalization of the observed phenomenon in the discussion. In addition, we will also change how we refer to it in introduction, on page 8 of Sec 3.2, and generally throughout the text of the paper.

"- I found some of the discussions regarding fragmentation hard to follow. For example, while Eq. (8) is clear enough, I would not have been able to write it based on the description that precedes it."

We will attempt to reformulate the paragraph before Eq (8) in order to make the reasoning clearer and easier to follow.

Attachment:

Figures.pdf

Anonymous on 2023-04-19  [id 3599]

(in reply to Pietro Brighi on 2023-03-08 [id 3452])
Category:
remark
question

As a small additional comment: a Hilbert space size of 3*10^8 still seems rather large for exact simulations. It is larger than the Hilbert space of N=28 spin-1/2 particles; in my experience, N=24-25 is around the largest one can do with reasonable numerical resources, even for fairly sparse Hamiltonians (i.e., only nearest-neighbor interactions). Were there any additional numerical tricks involved in these simulations?

Anonymous on 2023-04-21  [id 3604]

(in reply to Anonymous Comment on 2023-04-19 [id 3599])

Indeed, as the Referee correctly states, the Hilbert space dimension for L=37 is too large for exact simulations. Therefore, we used the 4th order Runge-Kutta method to numerically perform the time-evolution. This approximates the exact time-evolution $\psi(t)=e^{-i \hat{H}t}\psi_0$ with the sum of few matrix-vector multiplications, at the cost of an error scaling as $O(dt^4)$ (details about the time-step dt are given in the manuscript). The Runge-Kutta method then requires to store a few copies of vectors of $3\times 10^8$ double precision complex numbers, amounting each to roughly 5 GB of memory. This is still perfectly manageable by a powerful modern laptop or, in our case, a large memory desktop.

---

## Editorial Decision

published